# Continually Adding New Languages to Multilingual Language Models

**Abraham Toluwase Owodunni**                                              *owodunni.1@osu.edu*
*Department of Computer Science and Engineering*
*The Ohio State University*

**Sachin Kumar**                                                          *kumar.1145@osu.edu*
*Department of Computer Science and Engineering*
*The Ohio State University*

**Reviewed on OpenReview:** *https://openreview.net/forum?id=XXXX*

## Abstract

Multilingual language models are trained on a fixed set of languages, and to support new languages, the models need to be retrained from scratch. This is an expensive endeavor and is often infeasible, as model developers tend not to release their pre-training data. Naive approaches, such as continued pretraining, suffer from catastrophic forgetting; however, mitigation strategies like experience replay cannot be applied due to the lack of original pretraining data. In this work, we investigate the problem of continually adding new languages to a multilingual model, assuming access to pretraining data in only the target languages. We explore multiple approaches to address this problem and propose Layer-Selective LoRA (LAYRA), which adds Low-Rank Adapters (LoRA) to selected initial and final layers while keeping the rest of the model frozen. LAYRA builds on two insights: (1) LoRA reduces forgetting, and (2) multilingual models encode inputs in the source language in the initial layers, reason in English in intermediate layers, and translate back to the source language in final layers. We experiment with adding multiple combinations of Galician, Swahili, and Urdu to pretrained language models and evaluate each method on diverse multilingual tasks. We find that LAYRA provides the overall best tradeoff between preserving models' capabilities in previously supported languages, while being competitive with existing approaches such as LoRA in learning new languages. We also demonstrate that using model arithmetic, the adapted models can be equipped with strong instruction following abilities without access to any instruction tuning data in the target languages.

## 1 Introduction

Although several recently released language models (LMs) are advertised as multilingual (Grattafiori et al., 2024; Faysse et al., 2024; Gemma, 2025), they only support a handful of predetermined high-resource languages. As resources for new languages become available, continually supporting them in such models is not trivial. Retraining them from scratch is often prohibitively expensive, so practitioners typically adopt an incremental continued pretraining (CPT) strategy to incorporate new languages (Csaki et al., 2023). However, it often results in catastrophic forgetting of previously supported languages (Cahyawijaya et al., 2023; Chalkidis et al., 2021; Vu et al., 2022).

The most common solution to avoid forgetting is experience replay—reintroducing data in previously supported languages during CPT (Winata et al., 2023; Wang et al., 2024b). Unfortunately, most recent model releases are not accompanied by their pretraining data (Yang et al., 2025; Meta, 2025; Cohere et al., 2025; Rastogi et al., 2025; Zhao et al., 2025; Gemma, 2025). Even if the data were available or approximated using public sources, as the number of supported languages in an LM grows, replaying data in all of them can also become

computationally infeasible. Recent works proposing alternative approaches to mitigate forgetting have also been shown to work well only in conjunction with replay (Winata et al., 2023; Alexandrov et al., 2024; Chen et al., 2023; Aggarwal et al., 2024).

In this work, we study replay-free continual learning methods for adding new languages to multilingual LMs. We begin with two lightweight parameter-efficient strategies that have shown promise: updating only selected layers of the LM (Remy et al., 2024) and continued pretraining with Low-rank Adapters (LoRA) (Hu et al., 2022; Biderman et al., 2024a). While effective in some cases, we find that both approaches still suffer from significant forgetting. To address these limitations, we introduce LAYRA, which inserts LoRA modules into *selected* transformer layers in an LM during training, while keeping the rest of the model frozen. LAYRA is motivated by two observations: (1) LoRA can reduce forgetting but often underlearns (Biderman et al., 2024a), and (2) multilingual LMs process an input sequence in three stages as shown by Zhao et al. (2024) and Wendler et al. (2024). Using logit lens-based analysis (Nostalgebraist, 2020), they demonstrated that the earliest layers of the LM process the sequence in the language in which it is written, the middle layers process the sequence in English (the most dominant language in the pretraining corpora), and the final layers translate back and generate a response in the input language. By targeting only the layers responsible for handling non-English text, we find that LAYRA achieves stronger learning with further reduced forgetting.

We further show that by combining LAYRA with model merging methods, we can sequentially continue to add new languages to a pretrained LM. Finally, we show that we can enable instruction following capabilities of the adapted model in the new languages using instruction residuals extracted from aligned models in the previously supported languages.

We validate our method by adding different combinations of three typologically different languages with limited pretraining resources (Galician, Urdu, Swahili) to Llama 3.1 (Grattafiori et al., 2024) and Qwen 2.5 (Qwen-Team, 2024) (§3). We choose these languages to understand the impact of the writing script and the relatedness of target languages with the original model—on both learning the target language and not forgetting (i.e., retention) of already supported languages. Our results (§4) show that while CPT and LoRA are more efficient with learning new languages, LAYRA shines in retention while being competitive in learning new languages. We find that a new language, irrespective of its relatedness to the originally supported languages, can be added successfully, as long as its writing script is represented by the model. Finally, upon adding instruction vectors extracted from an existing instruction-tuned model, our adapted model acquires instruction-following abilities even without needing any language-specific instruction-tuning training. Since language models are only truly valuable to end users when they can follow instructions, we believe our findings on instruction-following will enable broader adaptation of these language models by speakers of low-resource languages.

## 2 The Multilingual Continual Learning Problem

### 2.1 Problem Setup

Suppose we have a pretrained autoregressive LM $\theta_N$ that supports $N > 1$ languages. Given pretraining data in $n$ new languages, $\{L_1, L_2, \ldots, L_n\}$, our goal is to create a model $\theta_{N+n}$ that supports all $N + n$ languages. Crucially, $\theta_{N+n}$ should retain its performance in the original $N$ languages (retention) while acquiring competence in the new $n$ languages (learning).

We also consider a generalized continual learning setup where given $\theta_{N+n}$, we update it to include $n'$ more languages, thus creating $\theta_{N+n+n'}$. In principle, this process can go on indefinitely as resources for new languages emerge, reflecting common practice in language modeling and machine learning, where new training data arrive incrementally (BLOOM; Leong et al., 2022), (Wura; Oladipo et al., 2023), or (FineWeb 2; Penedo et al., 2024)).

Furthermore, we assume no access to the pretraining data in the original $N$ languages that led to the creation of $\theta_N$. This also reflects a new reality in open-weights release of language models, where the pretraining data or its constitution is often not publicly shared by the organizations building them (Dubey et al., 2024; Bai et al., 2023; Abdin et al., 2024).

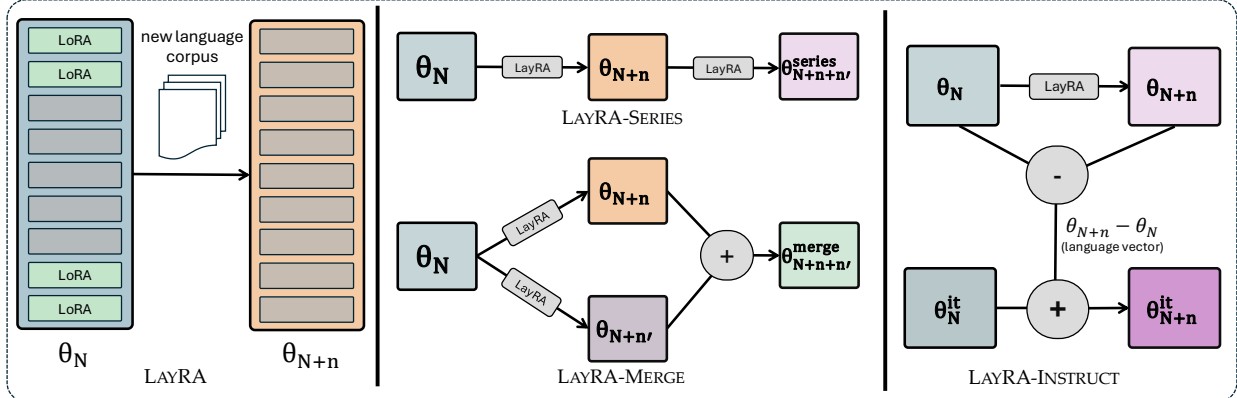

Figure 1: Problem setup for continually adding new languages and instruction tuning to a model. Left: Layer-selective LoRA, which performs LoRA updates to only selected initial and final transformer layers. Middle: Continual learning techniques to enable the addition of multiple languages. LAYRA-SERIES denotes the sequential addition of multiple languages, while LAYRA-PARALLEL denotes the addition of multiple languages with a single stage. Right: Enable instruction following in the adapted model without instruction data using model arithmetic. We note that this scheme can be used with all the continual pretraining methods we explored.

**Supporting instruction following in new languages**   To create models that can respond to user queries, pretrained LMs typically go through an instruction tuning phase using curated labeled datasets (Ouyang et al., 2022; Chung et al., 2024; Lambert et al., 2024). However, instruction-tuning datasets remain scarce or unavailable for most non-English languages. Hence, we explore data-free methods to add instruction following abilities to updated models $\theta_{N+n}$, assuming access to an instruction-tuned model that supports the original $N$ languages, $\theta_N^{\text{it}}$.

## 2.2   Method

We summarize our continual learning scheme in Figure 1. Our goal is to add new languages to $\theta_N$ without causing the model to forget the previously learned languages. The simplest and most naive approach to do this is to continue pretraining (CPT) $\theta_N$ with new data using a language modeling objective (such as next token prediction). However, it has been widely observed (and confirmed in our experiments) that CPT leads to severe catastrophic forgetting, causing large drops in performance on previously supported languages (Csaki et al., 2023). Experience replay, reintroducing past data during CPT, has been proposed as a viable approach to address this issue, but it requires access to pretraining data of $\theta_N$, which is not available to us. While multilingual pretraining datasets in many languages have recently been open-sourced, recent state-of-the-art multilingual LLMs are trained with such a large number of languages that it can become extremely computationally expensive to perform replay. For example, BloomZ (Muennighoff et al., 2022), Aya 23 (Aryabumi et al., 2024), and Gemma 3 (Gemma, 2025) were trained on 23, 46, and over 140 languages, respectively. This number is likely to grow over time.

Instead of experience replay, in this work, we explore continued pretraining with parameter-efficient approaches, such as LoRA (Hu et al., 2022), which have been shown in prior work to "learn less and forget less" (Biderman et al., 2024a), thus finding a better balance between learning new tasks and retaining past knowledge. Our experiments on adding languages also reveal similar findings. To further minimize forgetting and improve learning, we also explore the following improvements to the training procedure, updating only a subset of the layers during CPT.

**Layer-selective Continual Learning**   Recent work has indicated that multilingual LMs with layers $\{\mathcal{T}_l\}_{l=1}^{L}$ process sequences in three main stages (Zhao et al., 2024; Wendler et al., 2024).

1. The **earliest layers** ($\mathcal{T}_{\sim 1}$) encode the input in its source language.

2. The **middle layers** ($\mathcal{T}_{\sim L/2}$) handle the model's internal "reasoning language" (often English in models such as the Llama series of models).

3. The **final layers** ($\mathcal{T}_{\sim L}$) convert the representation back into the target language during generation.

Based on this observation, we hypothesize that training only the layers responsible for handling non-English text and freezing the English-specific layers should be sufficient to support new languages to the model, preserving its core reasoning abilities. We combine this layer-selective training method with parameter-efficient updates to propose our final training approach, which we call **Lay**er-selective Lo**RA** (**LayRA**). [1]

Given $n$ new languages, we first continue pretraining $\theta_N$ to obtain an adapted model $\phi_{N+n}$. We then obtain $\theta_{N+n}$ as,

$$\theta_{N+n} = \theta_N + \lambda(\phi_{N+n} - \theta_N) \tag{1}$$

Here $\lambda$ is a hyperparameter that controls the weights of the added language vector ($\phi_{N+n} - \theta_N$) that is learned during training with one or multiple languages. Similar to its usage in previous works (Morrison et al., 2024; Wang et al., 2024a), the addition here is a vector operation, and this helps to obtain the right balance of language vectors to add when working with multiple language vectors. The language vector in this equation could also be swapped for a task vector, as later seen in Equation 4. By setting $\lambda = 1$, this becomes equivalent to full model training with any chosen CPT method.

**Sequentially adding new languages over multiple stages** To support multiple stages of continual learning, where $n$ languages are added in the first stage, and $n'$ languages are added in the second stage (and so on), we explore the following two methods.

In the first setup, which we call LAYRA-SERIES, we iteratively apply LAYRA by first training $\theta_N$ to create $\theta_{N+n}$. We then continue training $\theta_{N+n}$ on $n'$ newer languages again following LAYRA creating

$$\theta_{N+n+n'}^{\text{series}} = \theta_{N+n} + \lambda'(\phi_{N+n+n'} - \theta_{N+n}). \tag{2}$$

Here, $\lambda'$ is another hyperparameter tuned separately from $\lambda$. This process can be continued indefinitely to add more languages. In the second setup, which we call LAYRA-MERGE, we first create $\theta_{N+n'}$ separately without relying on $\theta_{N+n}$ by applying LAYRA with the $n'$ languages on $\theta_N$. We then merge the weights of these specialized models, $\theta_{N+n}$ and $\theta_{N+n'}$, to yield a single model $\theta_{N+n+n'}$. Concretely,

$$\theta_{N+n+n'}^{\text{merge}} = \mu\theta_{N+n} + (1-\mu)\theta_{N+n'} \tag{3}$$

This approach aims to combine gains for each set of training without introducing additional forgetting issues from a previous LAYRA stage as in series. In practice, a practitioner may use different combinations of SERIES and MERGE depending on the languages being added to the underlying model.

**Adding Instruction Following Capabilities** So far, we have described an approach to add new languages to a pretrained model using raw text available in target languages. Recent open-weights models (Dubey et al., 2024; Bai et al., 2023; Abdin et al., 2024; OLMo et al., 2024) all follow a pattern of releasing both a pretrained (base) and an instruction-tuned model ($\theta_N^{\text{it}}$). To add instruction following abilities in the adapted model $\theta_{N+n}$ without any labeled data in the $n$ new languages, we compute a *language vector* as the difference between $\theta_{N+n}$ and $\theta_N$ and apply it to $\theta_N^{\text{it}}$ as,

$$\theta_{N+n}^{\text{it}} = \gamma(\theta_{N+n} - \theta_N) + \theta_N^{\text{it}} \tag{4}$$

By doing so, we inherit the instruction-following capabilities learned by $\theta_N^{\text{it}}$ while supporting newly added languages to create LAYRA-INSTRUCT.[2] The scaling factor $\gamma$ can be tuned to balance instruction performance

---

[1] A variation of this method, as evaluated by Remy et al. (2024), showed its validity in faster adaptation to low-resource languages. However, their method still suffered from catastrophic forgetting, which we aim to address in this work.

[2] While more sophisticated methods of model merging have recently been developed such as TIES (Yadav et al., 2023) and DARE (Yu et al., 2024), our initial experiments did not show improvements with them over task vectors.

and new-language retention. This method was inspired by the work on task arithmetic from Ilharco et al. (2022). Given instruction-tuning datasets in the target language, this model can further be improved, but we do not assume any such access in this work.

# 3 Experimental Setup

## 3.1 Languages, Datasets, and Model

We use Llama 3.1 8B (Grattafiori et al., 2024), which supports $N = 8$ languages for the majority of our experiments, and we tested our most optimal methods on Qwen 2.5 7B (Qwen-Team, 2024) to show that they generalize to other models. Six languages in Llama 3.1 8B use Latin script (English, German, French, Italian, Portuguese, and Spanish), while two use non-Latin scripts (Hindi in Devanagari and Thai in Thai script). The number of languages supported by Qwen 2.5 7B is not publicly known.

**New languages**   To test the impact of writing script and their relatedness to existing languages in the model, we experimented with adding the following languages:

- **Galician**: a mid-resource Romance language (Latin script) spoken in northwestern Spain. Given its similarities to Portuguese and Spanish (both of which exist in Llama 3.1), Galician is well-suited for leveraging prior multilingual knowledge.

- **Swahili**: a low-resource Bantu language predominantly spoken in East Africa by roughly 100 million speakers. It is not related to any of the languages in Llama 3.1 but is written in Latin script, which is well represented in the model.

- **Urdu**: a low-resource Indo-Aryan language (Perso-Arabic script). Although Urdu shares substantial linguistic commonalities with Hindi (which is supported by the Llama 3.1), its script differs from Hindi.

This diverse selection of languages provides a robust test of how effectively the model can learn distinct scripts and linguistic structures. For each of the three languages, we create our pretraining datasets using FineWeb-2 (Penedo et al., 2024). We use all available data for Swahili, which was ∼1.2B tokens. For the other two languages, we subsampled the corpus to contain the same number of tokens to control for the impact of dataset size. We chose FineWeb-2 because the dataset has been rigorously cleaned and decontaminated, leading to the removal of all benchmarking datasets. This cleaning pipeline is well outlined in the dataset paper (Penedo et al., 2024).

We conduct two sets of experiments: (1) a single-stage continual learning setup with $n = 1$ where we add only one of the three languages at a time to the base model, and, (2) a two-stage setup with $n = 1$ and $n' = 1$, where we first add one of the languages to the pretrained model, and incorporate a second language later on.

## 3.2 Hyperparameters

For our single-stage experiments (Equation 1), we set $\lambda = 1$, which adds the entire language vector that is obtained after CPT. This is analogous to a straightforward LoRA CPT (with selected layers), and we merge the weights of the LoRA modules with the pretrained to maintain consistent model sizes across all our experiments. In LAYRA-SERIES where we iteratively add more languages following Equation 2, we empirically determine $\lambda' = 0.5$ to perform the best[3]. Note that $\lambda = 1$ is the hyperparameter for adding a single language, while $\lambda' = 0.5$ is used for adding multiple languages in our LAYRA-SERIES setup.

In our second setup for adding multiple languages via merging, LAYRA-MERGE, we set $\mu = 0.5$, which is analogous to averaging all the adapted models from CPT. To add instruction following abilities to the adapted model as in Equation 4, we use a value of $\gamma = 0.7$, which adds part of the language vector to Llama 3.1 Instruct. We determine the value of $\gamma$ with a small-scale experiment. For the adapters, we use a rank ($r$)

---

[3]While we perform experiments with only two stages, future stages may require an even smaller multiplier

of 8 and $\alpha$ of 16. We use LoRA dropout of 0.05. We use this setup following Biderman et al. (2024a), which shows that these hyperparameters result in the least forgetting. For all LAYRA experiments, we apply LoRA to the earliest 10 and the final 2 layers. In addition, we also finetune the embedding layer and the LM head. We provide analysis and ablation studies that provide the reasoning for choosing these hyperparameters in §5. All other training hyperparameters can be found in Appendix A.4 Table 23.

### 3.3 Methods

In addition to LAYRA, we experimented with the following methods.

- **Full CPT** In this method, we continue pretraining all the base model parameters.

- **LoRA CPT** In this method, we continue pretraining the base model using low-rank adapters (LoRA) applied at all layers following (Biderman et al., 2024a).

- **Layer-Selective Full CPT** In this method, we fully train the first and the last transformer layer of the base model along with the embedding layer and the LM head (these layers were empirically determined to give the best average performance between retention and learning). This method also serves as an ablation of LAYRA with the adapters removed (see Tables 16, 17, 15, and 18 for our results on ablating layer selections. We note that this selection method could also be extended to include other, more sophisticated selection heuristics, but we leave that for our future work.

### 3.4 Evaluation

We evaluate the adapted pretrained models on XNLI (Natural Language Inference; Conneau et al., 2018), PAWS-X (Paraphrasing; Yang et al., 2019), XCOPA (Commonsense Reasoning; Ponti et al., 2020), and XStoryCloze (Commonsense Reasoning; Lin et al., 2021). We evaluate the instruction adapted models on XNLI, MGSM (Math; Shi et al., 2022), and MMLU-Lite (MCQs; Singh et al., 2024). For MGSM and MMLU, we generate the answer by greedily decoding from the model. We evaluate the rest as classification tasks by choosing the labels with the highest probability. We use the LM harness evaluation framework (Biderman et al., 2024b) for our evaluations. We use 0-shot evaluation for all tasks except for MGSM, for which we use a 3-shot setup following prior work (Rakuten-Group et al., 2024). Not all languages with which we experiment have datasets available for all tasks. For languages for which datasets are not available, we translate the English subset of the task to the missing language using Google Machine Translate (March 2025) (see Table 24 in Appendix A.4 for languages that required translations). We perform qualitative analysis to ensure that the translations are accurate. We do this by randomly sampling translated sentences, translate them back to English and assess if the back-translation preserves the original meaning in English. These datasets have been released on HuggingFace[4]. For all tasks, we report accuracy. For each task, we track *retention* measured by the lack of drop in performance in the originally supported languages and *learning*, which is measured by improvement in performance in the newly added language(s). An ideal solution leads to the highest retention while maximizing learning.

## 4 Results

### 4.1 Adding One Language to the Pretrained Model

We provide results for one-stage continual learning by adding one language at a time in Table 1 and Figure 2a. As expected, full CPT consistently exhibits the highest level of catastrophic forgetting across all languages and tasks—regardless of script, resource availability, or linguistic similarity of the target language to previously supported languages. Layer-Selective full CPT, by freezing most of the model layers and finetuning only the top and bottom layers, improves the learning-forgetting tradeoff. However, significant forgetting still persists. LoRA CPT, with its lightweight parameter updates, closes the gap even further. Our proposed approach, LayRA, considerably outperforms LoRA in terms of forgetting while being competitive with LoRA

---

[4] https://huggingface.co/collections/0wos/layra

Table 1: Performance of different CPT methods across languages for XNLI, PAWS-X, XStoryCloze, and XCOPA with Llama 3.1. See Tables 3, 4, 5 & 6 in the Appendix A.1 for full results with more languages, which we use to compute the average.

| | XNLI evaluation for models trained on Swa/Urd/Glg | | | | | | |
|---|---|---|---|---|---|---|---|
| **Model** | **eng** | **spa** | **hin** | **swa** | **urd** | **glg** | **Avg** |
| PT | 54.90 | 51.33 | 48.96 | 39.24 | 36.43 | 47.57 | 47.55 |
| Full | 52.93/49.88/50.88 | 44.42/34.86/47.31 | 34.18/35.82/33.90 | 45.46/34.26/32.61 | 33.45/39.68/37.43 | 37.19/34.26/50.93 | 40.95/37.57/41.21 |
| Layer-sel. | 55.06/54.70/53.01 | 45.50/36.27/47.11 | 41.93/39.08/37.83 | 46.71/35.06/**36.14** | 33.45/42.49/34.50 | 41.94/37.09/53.82 | 43.91/39.34/43.40 |
| LoRA | **55.90**/55.78/54.34 | **48.84**/42.89/49.32 | 45.30/39.12/45.70 | **47.71**/36.10/35.90 | **36.55**/**42.65**/36.75 | 44.08/**48.55**/**54.02** | **46.82**/44.82/46.49 |
| LAYRA | 54.22/**57.11**/**55.81** | 47.43/**43.82**/**50.64** | **46.83**/**42.01**/**48.07** | 45.34/**36.83**/34.66 | 34.98/40.96/**37.55** | **45.48**/47.81/**54.02** | 46.64/**45.58**/**47.14** |

| | PAWS-X evaluation for models trained on Swa/Urd/Glg | | | | | | |
|---|---|---|---|---|---|---|---|
| **Model** | **eng** | **spa** | **hin** | **swa** | **urd** | **glg** | **Avg** |
| PT | 67.45 | 65.30 | 64.45 | 61.00 | 54.25 | 63.55 | 63.17 |
| Full | 65.25/66.60/61.70 | 61.50/54.90/62.65 | 61.50/52.95/54.15 | **63.00**/52.95/53.35 | 55.05/46.85/47.50 | 50.80/47.15/62.85 | 59.36/54.11/57.37 |
| Layer-sel. | 64.65/66.95/58.90 | 60.60/58.40/63.95 | 59.00/54.35/56.05 | 60.60/49.65/51.00 | 55.05/52.20/49.10 | 55.70/53.60/**67.80** | 59.82/56.10/58.57 |
| LoRA | 68.75/**70.45**/67.15 | **64.00**/61.10/**64.60** | 63.35/**64.30**/62.85 | 61.70/48.00/46.15 | **59.90**/49.35/**50.45** | 56.70/59.10/66.60 | 62.97/59.58/60.89 |
| LAYRA | **68.95**/68.45/**67.15** | 63.85/60.35/64.30 | **64.20**/63.70/**63.30** | 61.20/**53.50**/50.35 | 59.75/**54.85**/49.45 | **59.65**/**60.80**/65.25 | **63.25**/**60.79**/**61.03** |

| | XCOPA evaluation for models trained on Swa/Urd/Glg | | | | | | |
|---|---|---|---|---|---|---|---|
| **Model** | **eng** | **spa** | **tha** | **swa** | **urd** | **glg** | **Avg** |
| PT | 87.00 | 81.40 | 57.60 | 55.00 | 58.80 | 57.60 | 67.14 |
| Full | 72.00/73.00/77.00 | 57.40/50.20/70.20 | 55.60/52.40/55.00 | 66.80/53.60/53.60 | 53.40/59.60/54.80 | 53.40/50.80/59.00 | 58.97/56.09/60.09 |
| Layer-sel. | 83.00/77.00/80.00 | 60.20/56.40/76.20 | 57.60/54.20/56.60 | 66.00/**54.00**/53.40 | 53.20/57.60/57.20 | 54.00/53.40/58.00 | 61.00/58.37/62.34 |
| LoRA | **88.00**/**86.00**/85.00 | **70.60**/74.80/**77.00** | 56.60/58.00/55.40 | 66.20/53.60/53.00 | 53.80/59.40/**59.80** | 56.00/58.00/62.20 | **64.74**/65.57/65.14 |
| LAYRA | 87.00/**86.00**/86.00 | 69.60/**76.80**/76.80 | **57.80**/**60.60**/58.40 | 64.60/53.40/**54.40** | **56.00**/**61.00**/57.40 | 52.40/56.00/**63.20** | 64.09/**66.26**/**65.20** |

| | XStoryCloze evaluation for models trained on Swa/Urd/Glg | | | | | | |
|---|---|---|---|---|---|---|---|
| **Model** | **eng** | **spa** | **hin** | **swa** | **-** | **glg** | **Avg** |
| PT | 78.16 | 70.75 | 64.46 | 55.86 | - | 64.46 | 66.74 |
| Full | 70.22/69.09 | 59.03/65.39 | 47.12/48.91 | 47.12/48.38 | - | 64.33/68.56 | 57.56/60.07 |
| Layer-sel. | 76.57/75.84 | 66.05/69.49 | 55.46/52.61 | 64.99/49.90 | - | 54.47/70.42 | 63.51/63.65 |
| LoRA | 76.17/**76.77** | **66.71**/69.82 | **63.40**/63.67 | 57.91/**51.42** | - | **65.12**/70.81 | **65.86**/66.50 |
| LAYRA | **76.51**/76.11 | 66.51/69.69 | 63.27/63.20 | **63.73**/50.69 | - | 57.91/69.89 | 65.59/65.92 |

in terms of learning across all tasks and target languages. These observations hold for both Llama and Qwen models. We provide detailed results across multiple tasks and additional languages for both models in Tables 3, 4, 5, 6, 7, 8, 9, and 10, which are included in Appendix A.1.

In these results, LAYRA achieves an average improvement of over 1 point compared to all other methods on 12 of the 19 groups of experiments. This gain is consistent across multiple LAYRA models trained on different languages and evaluated on our chosen benchmarks.

With LAYRA, we see the least amount of forgetting in English across all tasks, which is the anchor language. This aligns with our hypothesis that freezing the model's middle transformer layers preserves the core capabilities encoded in the anchor language.

**Impact of language relatedness and scripts** Using LAYRA, the Galician-trained model exhibits the strongest retention-learning tradeoff (see Table 1). We hypothesize that this result is due to positive transfer from Spanish and Portuguese (which are both supported by Llama 3.1). In fact, this model improves English performance across multiple tasks, highlighting that cross-lingual transfer may happen in both directions with our proposed approach. Swahili, while unrelated to any of the languages in the original model, also responds well to our training strategy with a substantial performance gain and a good amount of retention (albeit slightly worse than Galician) (Tables 3, 4, 5, 6). We speculate that this result is due to Latin script being well supported in the original model as well as the heavy usage of English loanwords in Swahili (Martin et al., 2021). Llama 3.1 models trained with Urdu, which is very closely related to Hindi, achieve the poorest overall performance, both in terms of learning and retention. Whereas the Qwen 2.5 models trained with Urdu exhibit a higher retention rate when compared to training on Swahili (Table 8, PAWS-X; Table 10, XCOPA; Table 9, XStoryCloze). We attribute this result to Urdu's writing script not being well represented in the Llama 3.1 model's tokenizer, leading to overfragmentation and poor adaptation. Qwen 2.5, on the other hand, supports Arabic, which has a similar writing script to Urdu.

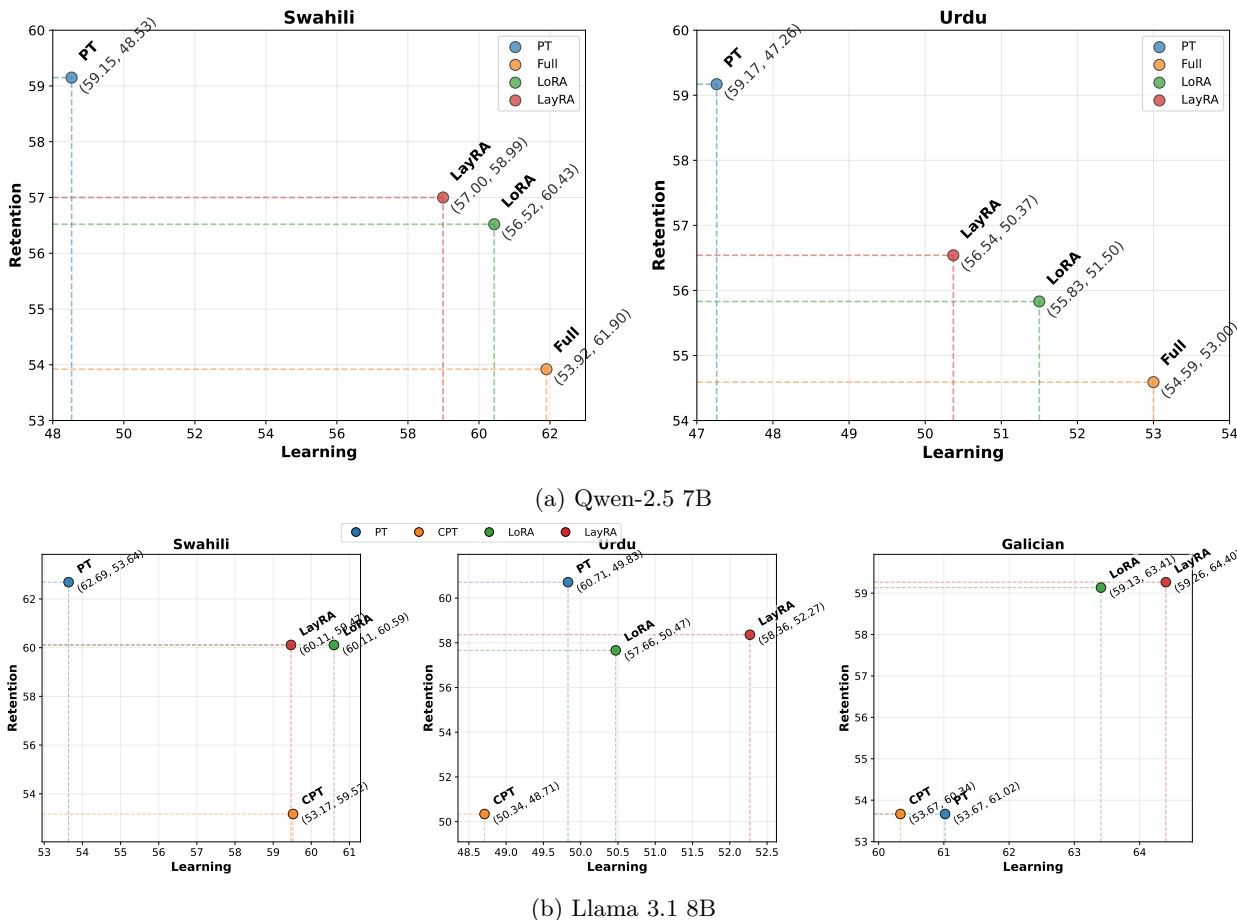

(a) Qwen-2.5 7B

(b) Llama 3.1 8B

Figure 2: Average accuracy of retention vs learning (y, x) on XNLI, PAWS-X, XCOPA, and XStoryCloze. We compute the average learning score (x) as the model's average score on the single language we trained on, while the retention score (y) as the average across all other 8 languages we worked with. See Tables 7–10 in Appendix A.1 for full Qwen results and Table 1 for full Llama results.

Although prior work has sought to address such issues of writing script with vocabulary expansion techniques (Kim et al., 2024), the resulting change in the number of model parameters hinders the use of model merging or parameter-efficient techniques to reduce catastrophic forgetting. Another work has shown that Tokenizer-free models may be a viable future direction in addressing this issue (Ahia et al., 2024; Owodunni et al., 2025). Furthermore, we observe that non-Latin-scripted languages such as Thai and Hindi also disproportionately suffer at retention across all methods, highlighting a broader trend of negative transfer between languages that do not share scripts. Others (Abagyan et al., 2025) have used tokenizers with very large vocabulary size (Global Tokenizers) to address this issue, although this comes with the overhead of more parameters in the model. Due to these observations, we exclude Urdu from further experiments and only report results with Galician and Swahili for two-stage continual learning.

## 4.2 Sequentially Adding Multiple Languages to the Pretrained Model

We provide the results for sequential addition of Galician and Swahili, LAYRA-SERIES and -MERGE in Table 2. Both of these methods assume that the resources of the languages arrived in order and not at the same time. For reference, we also include results for CPT assuming datasets for both languages were indeed available at the same time (referred to as PARALLEL). Unsurprisingly, PARALLEL produced the highest overall learning-retention trade-off, indicating the effectiveness of single-stage adaptation with multiple languages

Table 2: Performance of different LAYRA setups for adding two languages (Galician + Swahili) across XNLI, PAWS-X, XStoryCloze and XCOPA. Top for XNLI & PAWS-X. Bottom for XStoryCloze & XCOPA. See Table 11, 13, 12 & 14 for full results with more languages.

| Tasks | XNLI / PAWS-X | | | | | | |
|---|---|---|---|---|---|---|---|
| Model | deu | eng | spa | fra | swa | glg | Avg |
| PT | 52.05/66.20 | 54.90/67.45 | 51.33/65.30 | 50.12/64.45 | 39.24/61.00 | 47.57/63.55 | 48.95/64.66 |
| Parallel | 50.00/66.25 | 53.53/66.05 | 50.32/64.65 | 45.94/62.25 | 43.73/58.05 | **56.08**/**67.35** | 49.11/64.10 |
| Series (Glg→Swa) | 48.88/65.10 | 54.74/66.80 | 50.96/62.90 | 48.63/63.50 | **43.86**/53.75 | 52.01/55.65 | 48.98/61.28 |
| Series (Swa→Glg) | **51.81**/64.10 | **56.06**/**67.35** | 51.24/64.55 | 48.11/**65.45** | 42.97/47.50 | 54.82/64.20 | **49.51**/62.19 |
| Merging | **51.00**/**67.30** | 54.50/**67.45** | **51.81**/**65.30** | 49.92/64.45 | 43.57/**61.00** | 53.98/63.55 | 49.00/**64.66** |
| Tasks | XStoryCloze / XCOPA | | | | | | |
| Model | eng | spa | hin/tha | - | swa | glg | Avg |
| PT | 78.16/87.00 | 70.75/81.40 | 64.46/57.60 | - | 55.86/55.00 | 64.46/57.60 | 66.74/68.53 |
| Parallel | **76.64**/86.00 | **69.16**/76.80 | 63.47/57.00 | | **62.14**/**63.40** | **69.95**/60.80 | **68.27**/67.87 |
| Series (Glg→Swa) | 75.91/81.00 | 68.23/74.40 | 64.13/57.40 | - | 58.44/59.40 | 67.31/**61.80** | 66.80/66.37 |
| Series (Swa→Glg) | 75.45/83.00 | 68.56/76.40 | 63.53/57.60 | - | 57.45/60.00 | 67.90/59.20 | 66.58/66.37 |
| Merging | **76.64**/**87.00** | 69.03/**78.80** | **64.26**/**58.00** | - | 56.45/58.20 | 66.51/60.20 | 66.58/**68.10** |

($n = 2$). This method serves as the upper bound for the multi-stage learning approaches. With LAYRA-SERIES, the gains tend to shift toward the most recently added language with a slight forgetting of previously acquired languages. Adding related languages at the second stage (Swahili, then Galician) leads to better retention. In comparison, LAYRA-MERGE performs much better matching or even surpassing PARALLEL, yielding the highest retention of knowledge for languages employing Latin scripts.

### 4.3 Adding Instruction Tuning to the Adapted Model

In Figure 3, we observe that adding an instruction residual (as described in Equation 4) can enable LAYRA and other adapted models to follow user instructions. XNLI shows clear trends of improvement of the base instruction model across all three languages in our experiments. With MGSM and MMLU, the trends are not consistent. For MGSM, we observe an increase in accuracy for Swahili and Urdu but a decline in performance for Galician. For MMLU, both Urdu and Galician show declines. While we do not identify clear reasons for this performance drop, we attribute it to tokenization issues with Urdu and previously identified issues with simple model arithmetic techniques (Yadav et al., 2023; Tao et al., 2024). Given a small amount of instruction tuning data in the target languages, this gap may be filled. We also measure how much our LAYRA-INSTRUCT models forget their previous knowledge by evaluating them on the English version of our mentioned tasks in Figure 4 in Appendix A.4. We find that our adapted models still achieve high accuracy in English and, at times, outperform both Llama and Qwen instruction models.

## 5 Ablations

**Varying the number of layers for LayRA.** To choose the optimal number of layers that balances the learning-forgetting tradeoff during CPT, we ranged the number of early and final transformer layers to be finetuned from 1 to 10 each. We conduct this evaluation with Swahili and find that the combination of the first 10 and last 2 layers gives us the best balance. See Table 15, 16, 17, and 18 in Appendix A.2 for details.

**Language vector scaling in LayRA-series.** We investigate the impact of changing the language vector added during the sequential addition of multiple languages. We continually increase $\lambda'$ (from Equation 2) for LAYRA-SERIES from 0 to 1 in our Galician and Swahili series experiment (Glg→Swa). For all the tasks we evaluated on, (see Tables 19, 20, 21, 22 in Appendix A.3), as $\lambda'$ tends to 1, we observe more retention of the Swahili and more forgetting of Galician while there is a general drop in accuracy for all the previously learned languages.

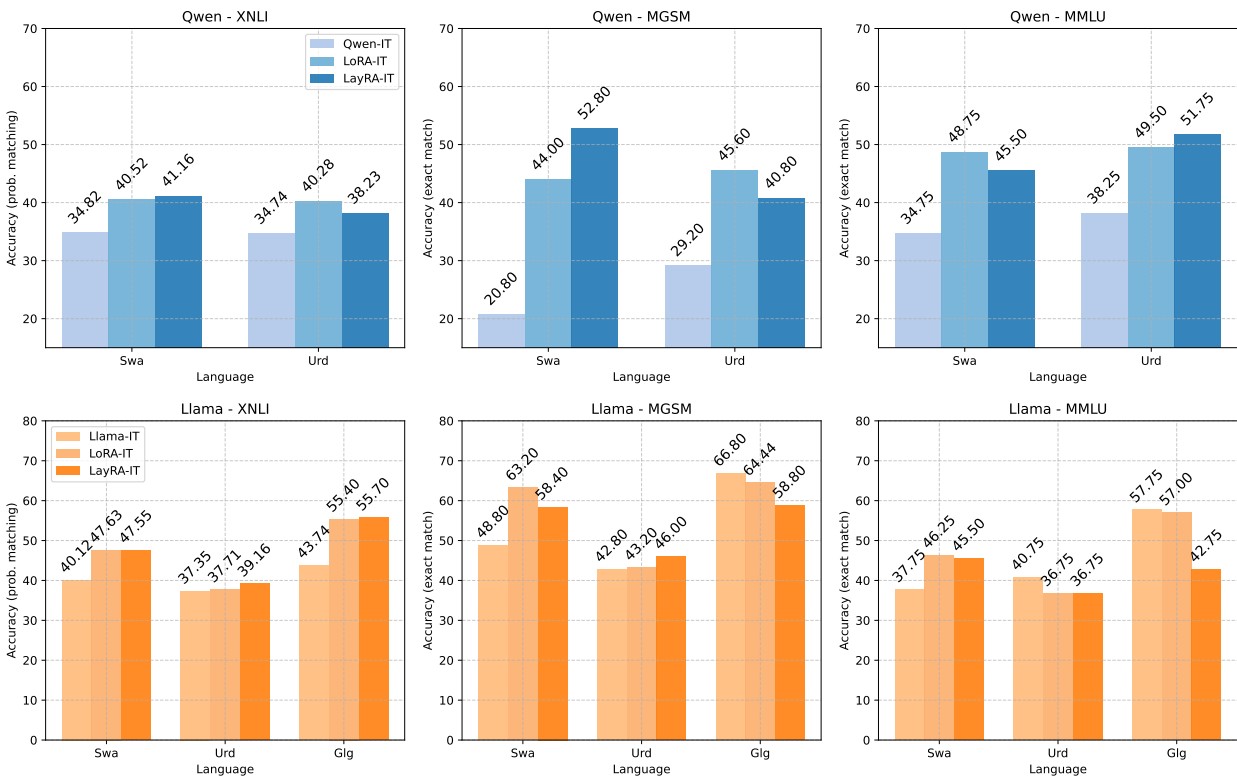

Figure 3: Accuracy of Qwen (top) and Llama (bottom) instruction (IT) models vs the adapted models on XNLI, MGSM, and MMLU.

## 6  Related Work

**Language Adaptation**  There exists extensive prior research to adapt LMs to new languages (Ogueji et al., 2021; Alabi et al., 2022; Lu et al., 2024). Most studies have focused on continued pretraining of all parameters of the models (Csaki et al., 2023; Alabi et al., 2022; Abagyan et al., 2025), adding new parameters such as adapters (Yong et al., 2022), or training a small subset of the model parameters (Pfeiffer et al., 2020; Houlsby et al., 2019; Remy et al., 2024). Similarly, our work uses adapters (LoRA) and applies them to a subset of model layers. These approaches are motivated by training LMs in the target languages(s), not preserving the performance in the original ones. They benefit from cross-lingual transfer of encoded knowledge in the pretrained models. If the script of the target knowledge is not supported by the pretrained models' tokenizer, Han et al. (2024) shows that adapting can be challenging. We demonstrate a similar issue with adapting Llama 3.1 to Urdu. A commonly proposed solution to address this issue is to expand vocabulary before continuing pretraining (Liu et al., 2023; Dobler & De Melo, 2023; Mundra et al., 2024). However, these techniques are known to exacerbate the forgetting issue (Mundra et al., 2024); model merging techniques to mitigate the issue cannot be applied due to different model sizes.

**Mitigating Catastrophic Forgetting**  Catastrophic forgetting is a well-known issue in neural models and remains a challenge for modern LMs, even for other cases beyond language adaptation. Reintroducing original data during adaptation (known as experience replay) is a commonly adopted remedy (Rolnick et al., 2019; Csaki et al., 2023; Winata et al., 2023). We explore strategies that do not assume access to the original data, which is a new reality in the case of modern LMs. Specifically, we modify LoRA by restricting it to select layers to improve this tradeoff. We leave exploration of learning rate schedules along with LAYRA for future work.

**Model Arithmetic** As a way to combine multiple models without training, model merging has been widely explored in the context of modern LMs (Hammoud et al., 2024; Dziadzio et al., 2024; Yang et al., 2024). Since its inception, many advanced merging techniques have been explored in recent works (Yadav et al., 2023; Yu et al., 2024; Kim et al., 2023). In our early exploration, they did not outperform the simplest arithmetic technique for creating task vectors proposed in Ilharco et al. (2022). Hence, we adopt it for sequential adaptation and for creating our instruction-adapted model. Multiple works have also explored model arithmetic during continued pretraining or fine-tuning, showing it can match or improve the performance of training from scratch. Most related to our work is BAM (Alexandrov et al., 2024), which performs full finetuning and merging after every few iterations, but they used experience replay, which is not applicable to our setup.

## 7 Conclusion

We studied the continual learning problem in a replay-free setting using multiple training methods. We also introduced new training schemes and method LAYRA, a layer-selective adapter-based method to continuously add new languages to a multilingual LLM. By strategically updating only the first and the last few transformer layers, we found that LAYRA effectively preserves knowledge of previously supported languages but learns the least when compared to LoRA and CPT. Our experiments demonstrate that this targeted, low-rank adaptation approach not only mitigates catastrophic forgetting but also benefits from cross-lingual transfer and can improve performance on existing languages. In addition, our merging strategies enable sequential continual learning, maintaining a favorable balance between stability and plasticity. Finally, we showed the potential of our adapted models to integrate instruction-following capabilities, even in scenarios where instruction-tuning data for newly added languages is not available. We tested our approach with two model families and multiple low-resource languages. We leave the exploration of model size and data for future work.

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

# A  Appendix

## A.1  Full results for LayRA

This section provides comprehensive tables detailing the complete set of experimental results for all methods across all languages and evaluation tasks (XNLI: Tables 3, 7; PAWS-X: Tables 4, 8; XCOPA: Tables 6, 9; XStoryCloze: Tables 5, 10).. These results offer further evidence supporting the conclusions drawn in the main text §4, allowing for deeper comparison and validation of performance metrics. Visual representations of these results are shown in Figure 2b and Figure 2a.

Table 3: Performance of different CPT methods across languages for XNLI with Llama 3.1

| Model | deu | eng | spa | fra | hin | tha | swa | urd | glg | Avg |
|---|---|---|---|---|---|---|---|---|---|---|
| PT | 52.05 | 54.90 | 51.33 | 50.12 | 48.96 | 47.39 | 39.24 | 36.43 | 47.57 | 47.55 |
| Swa Full | 41.33 | 52.93 | 44.42 | 44.54 | 34.18 | 35.02 | 45.46 | 33.45 | 37.19 | 40.95 |
| Swa Layer-sel. | 44.78 | 55.06 | 45.50 | 48.27 | 41.93 | 37.59 | 46.71 | 33.45 | 41.94 | 43.91 |
| Swa LoRA | 48.63 | **55.90** | **48.8**4 | 48.31 | 45.30 | 46.02 | 47.71 | **36.55** | 44.08 | **46.82** |
| Swa LayRA | **49.92** | 54.22 | **47.43** | 49.24 | 46.83 | 46.35 | 45.34 | 34.98 | **45.48** | 46.64 |
| Urd Full | 37.11 | 49.88 | 34.86 | 35.86 | 35.82 | 36.39 | 34.26 | 39.68 | 34.26 | 37.57 |
| Urd Layer-sel. | 37.91 | 54.70 | 36.27 | 36.87 | 39.08 | 34.58 | 35.06 | 42.49 | 37.09 | 39.34 |
| Urd LoRA | 46.18 | 55.78 | 42.89 | 49.32 | 39.12 | 42.77 | 36.10 | **42.65** | 48.55 | 44.82 |
| Urd LayRA | **48.39** | **57.11** | **43.82** | **50.00** | 42.01 | 43.29 | **36.83** | 40.96 | 47.81 | **45.58** |
| Glg Full | 43.17 | 50.88 | 47.31 | 40.84 | 33.90 | 33.82 | 32.61 | 37.43 | 50.93 | 41.21 |
| Glg Layer-sel. | 47.67 | 53.01 | 47.11 | 43.57 | 37.83 | 36.99 | **36.14** | 34.50 | 53.82 | 43.40 |
| Glg LoRA | **51.77** | 54.34 | 49.32 | 45.70 | 45.70 | 44.90 | 35.90 | 36.75 | **54.02** | 46.49 |
| Glg LayRA | 50.84 | **55.81** | **50.64** | **46.22** | **48.07** | **46.43** | 34.66 | **37.55** | **54.02** | **47.14** |

Table 4: Performance of different CPT methods across languages for PAWS-X with Llama 3.1

| Model | deu | eng | spa | fra | swa | urd | glg | Avg |
|---|---|---|---|---|---|---|---|---|
| PT | 66.20 | 67.45 | 65.30 | 64.45 | 61.00 | 54.25 | 63.55 | 63.17 |
| Swa Full | 58.45 | 65.25 | 61.50 | 61.50 | **63.00** | 55.05 | 50.80 | 59.36 |
| Swa Layer-sel. | **63.15** | 64.65 | 60.60 | 59.00 | 60.60 | 55.05 | 55.70 | 59.82 |
| Swa LoRA | **66.40** | 68.75 | **64.00** | 63.35 | 61.70 | **59.90** | 56.70 | 62.97 |
| Swa LayRA | 65.15 | **68.95** | 63.85 | **64.20** | 61.20 | 59.75 | **59.65** | **63.25** |
| Urd Full | 57.35 | 66.60 | 54.90 | 52.95 | 52.95 | 46.85 | 47.15 | 54.11 |
| Urd Layer-sel. | 57.55 | 66.95 | 58.40 | 54.35 | 49.65 | 52.20 | 53.60 | 56.10 |
| Urd LoRA | **64.75** | **70.45** | 61.10 | **64.30** | 48.00 | 49.35 | 59.10 | 59.58 |
| Urd LayRA | 63.90 | 68.45 | **60.35** | 63.70 | **53.50** | **54.85** | 60.80 | **60.79** |
| Glg Full | 59.40 | 61.70 | 62.65 | 54.15 | 53.35 | 47.50 | 62.85 | 57.37 |
| Glg Layer-sel. | 63.20 | 58.90 | 63.95 | 56.05 | 51.00 | 49.10 | **67.80** | 58.57 |
| Glg LoRA | **68.40** | **67.15** | **64.60** | 62.85 | 46.15 | 50.45 | 66.60 | 60.89 |
| Glg LayRA | 67.40 | **67.15** | 64.30 | **63.30** | **50.35** | 49.45 | 65.25 | **61.03** |

## A.2  LayRA layer selection ablation results

Here we present results for ablation studies done to find the best layer combinations for LayRA. The tables in this section contain all the languages and tasks we evaluate our models on. The tables are also split up and expanded for easier comprehension.

Table 5: Performance of different CPT methods across languages for XStoryCloze with Llama 3.1

| Model | eng | spa | hin | swa | glg | Avg |
|---|---|---|---|---|---|---|
| PT | 78.16 | 70.75 | 64.46 | 55.86 | 64.46 | 66.74 |
| Swa Full | 70.22 | 59.03 | 47.12 | 47.12 | 64.33 | 57.56 |
| Swa Layer-sel. | 76.57 | 66.05 | 55.46 | 64.99 | 54.47 | 63.51 |
| Swa LoRA | 76.17 | **66.71** | **63.40** | **57.91** | 65.12 | **65.86** |
| Swa LAYRA | **76.51** | 66.51 | 63.27 | 63.73 | 57.91 | 65.59 |
| Glg Full | 69.09 | 65.39 | 48.91 | 48.38 | 68.56 | 60.07 |
| Glg Layer-sel. | 75.84 | 69.49 | 52.61 | 49.90 | 70.42 | 63.65 |
| Glg LoRA | **76.77** | 69.82 | **63.67** | **51.42** | **70.81** | **66.50** |
| Glg LAYRA | 76.11 | **69.69** | 63.20 | 50.69 | 69.89 | 65.92 |

Table 6: Performance of different CPT methods across languages for XCOPA, with Llama 3.1

| Model | eng | spa | ita | tha | swa | urd | glg | Avg |
|---|---|---|---|---|---|---|---|---|
| PT | **87.00** | **81.40** | **72.60** | **57.60** | 55.00 | 58.80 | 57.60 | 67.14 |
| Swa Full | 72.00 | 57.40 | 54.20 | 55.60 | **66.80** | 53.40 | 53.40 | 58.97 |
| Swa Layer-sel. | 83.00 | 60.20 | 53.00 | 57.60 | 66.00 | 53.20 | 54.00 | 61.00 |
| Swa LoRA | **88.00** | **70.60** | **62.00** | 56.60 | 66.20 | 53.80 | **56.00** | 64.74 |
| Swa LAYRA | 87.00 | 69.60 | 61.20 | **57.80** | 64.60 | **56.00** | 52.40 | **64.09** |
| Urd Full | 73.00 | 50.20 | 53.00 | 52.40 | 53.60 | 59.60 | 50.80 | 56.09 |
| Urd Layer-sel. | 77.00 | 56.40 | 56.00 | 54.20 | **54.00** | 57.60 | 53.40 | 58.37 |
| Urd LoRA | **86.00** | 74.80 | 69.20 | 58.00 | 53.60 | 59.40 | **58.00** | 65.57 |
| Urd LAYRA | **86.00** | **76.80** | **70.00** | **60.60** | 53.40 | **61.00** | 56.00 | **66.26** |
| Glg Full | 77.00 | 70.20 | 51.00 | 55.00 | 53.60 | 54.80 | 59.00 | 60.09 |
| Glg Layer-sel. | 80.00 | 76.20 | 55.00 | 56.60 | 53.40 | 57.20 | 58.00 | 62.34 |
| Glg LoRA | **85.00** | 77.00 | **63.60** | 55.40 | 53.00 | **59.80** | 62.20 | 65.14 |
| Glg LAYRA | 86.00 | 76.80 | 60.20 | **58.40** | 54.40 | 57.40 | **63.20** | **65.20** |

## A.3 Changing the language vector in LayRA-series.

This appendix shows the sensitivity of the model to the scaling factor $\lambda'$ used when sequentially adding new languages with LAYRA-SERIES. We include detailed tables and analyses to illustrate how changing this hyperparameter influences retention of previously acquired languages versus performance gains on newly added ones.

## A.4 Other Tables and Figures

We put all other tables and figure in this section such as hyperparameter 23 table containing exhaustive details regarding experimental setups, including training hyperparameters such as learning rates, batch sizes, etc. We also add a table to show the languages we translate in Table 24 and Figure to show the forgetting rate of LAYRA-INSTRUCT on English (Figure 4).

Table 7: Performance of different CPT methods across languages for XNLI with Qwen 2.5

| Model | deu | eng | spa | fra | glg | hin | tha | swa | urd | Avg |
|---|---|---|---|---|---|---|---|---|---|---|
| PT | 47.38 | 54.01 | 48.83 | 50.88 | **48.97** | **43.25** | **44.61** | 34.77 | 34.73 | **45.27** |
| Swa Full | 41.52 | 53.21 | 45.86 | 43.05 | 37.17 | 34.29 | 34.65 | **46.50** | 33.93 | 41.13 |
| Swa LoRA | **47.95** | 53.21 | **49.51** | 49.71 | 37.49 | 39.03 | 40.08 | **46.86** | 34.13 | **44.22** |
| Swa LAYRA | 47.55 | **55.10** | 48.55 | **50.52** | **39.38** | **39.75** | 40.48 | 44.49 | **33.41** | 44.36 |
| Urd Full | 46.02 | 52.44 | 44.97 | 44.89 | 42.14 | **42.48** | 33.73 | 33.49 | **45.06** | 42.80 |
| Urd LoRA | **50.24** | 49.03 | 39.19 | **50.96** | **47.63** | 35.98 | 35.02 | 32.77 | 43.45 | 42.70 |
| Urd LAYRA | 48.63 | **51.88** | **48.75** | 49.79 | 45.83 | 37.51 | **37.59** | **35.30** | 40.36 | **43.96** |

Table 8: Performance of different CPT methods across languages for PAWS-X with Qwen 2.5

| Model | deu | eng | spa | fra | glg | swa | urd | Avg |
|---|---|---|---|---|---|---|---|---|
| PT | 64.40 | 69.85 | 65.50 | 66.45 | 59.35 | 54.75 | 53.45 | 61.96 |
| Swa Full | 58.25 | 69.85 | 64.10 | 62.50 | 50.50 | **64.75** | 52.95 | 60.41 |
| Swa Layer-sel. | 61.75 | 69.29 | 61.90 | 62.20 | 47.90 | 63.65 | 58.50 | 60.74 |
| Swa LoRA | 59.55 | 69.85 | 64.55 | 67.05 | 51.70 | 63.85 | 51.90 | 61.21 |
| Urd Full | 62.90 | 68.95 | 62.10 | 65.90 | 57.65 | 47.85 | **53.95** | 59.90 |
| Urd Layer-sel. | 61.60 | 71.30 | 64.65 | 60.55 | 61.65 | 47.15 | 53.45 | 60.05 |
| Urd LoRA | 57.70 | 71.00 | 66.45 | 57.90 | 54.50 | 46.80 | 51.75 | 58.01 |

Table 9: Performance of different CPT methods across languages for XCOPA with Qwen 2.5

| Model | eng | esp | glg | tha | swa | urd | Avg |
|---|---|---|---|---|---|---|---|
| PT | 90.00 | 79.20 | 54.40 | 74.40 | 52.80 | 53.60 | 66.09 |
| Swa Full | 78.00 | 66.00 | 50.60 | 58.80 | **69.79** | 54.40 | 61.46 |
| Swa LoRA | 80.00 | 71.60 | 55.40 | 62.00 | 66.60 | 54.60 | 64.00 |
| Swa LAYRA | 82.00 | 72.60 | 55.20 | 63.40 | 64.80 | 56.39 | **64.54** |
| Urd Full | 82.00 | 65.60 | 53.00 | 59.40 | 53.40 | **60.00** | 61.09 |
| Urd LoRA | 87.00 | 68.79 | 55.20 | 65.60 | 53.40 | 57.60 | 63.37 |
| Urd LAYRA | 87.00 | 74.20 | 56.59 | 67.80 | 52.40 | 59.00 | **65.06** |

Table 10: Performance of different CPT methods across languages for XStoryCloze with Qwen 2.5

| Model | eng | spa | hin | glg | swa | Avg |
|---|---|---|---|---|---|---|
| PT | 77.49 | 69.02 | 58.17 | 60.82 | 51.81 | 63.46 |
| Swa Full | 71.67 | 63.86 | 52.21 | 51.42 | 66.57 | 61.15 |
| Swa LoRA | 76.30 | 65.51 | 57.64 | 54.93 | 64.59 | 63.79 |
| Swa LAYRA | 76.50 | 65.32 | 57.84 | 56.12 | 62.80 | **63.72** |

Table 11: Performance of different LAYRA setups for adding two languages (Galician + Swahili) on XNLI

| Model | deu | eng | spa | fra | hin | tha | swa | glg | Avg |
|---|---|---|---|---|---|---|---|---|---|
| PT | 52.05 | 54.90 | 51.33 | 50.12 | 48.96 | 47.39 | 39.24 | 47.57 | 48.95 |
| Series (Glg→Swa) | 48.88 | 54.74 | 50.96 | 48.63 | 47.95 | **44.82** | **43.86** | 52.01 | 48.98 |
| Series (Swa→Glg) | **51.81** | **56.06** | 51.24 | 48.11 | 46.87 | 44.18 | 42.97 | 54.82 | 49.51 |
| Parallel | 50.00 | 53.53 | 50.32 | 45.94 | **48.63** | 44.66 | 43.73 | **56.08** | 49.11 |
| Merging | 51.00 | 54.50 | **51.81** | **49.92** | 44.86 | 42.33 | 43.57 | 53.98 | 49.00 |

Table 12: Performance of different LAYRA setups for adding two languages (Galician + Swahili) on XStoryCloze

| Model | eng | spa | hin | swa | glg | Avg |
|---|---|---|---|---|---|---|
| PT | 78.16 | 70.75 | 64.46 | 55.86 | 64.46 | 66.74 |
| Series (Glg→Swa) | 75.91 | 68.23 | 64.13 | 58.44 | 67.31 | 66.80 |
| Series (Swa→Glg) | 75.45 | 68.56 | 63.53 | 57.45 | 67.90 | 66.58 |
| Parallel | **76.64** | **69.16** | 63.47 | **62.14** | **69.95** | **68.27** |
| Merging | **76.64** | 69.03 | **64.26** | 56.45 | 66.51 | 66.58 |

Table 13: Performance of different LAYRA setups for adding two languages (Galician + Swahili) on PAWS-X

| Model | deu | eng | spa | fra | swa | glg | Avg |
|---|---|---|---|---|---|---|---|
| PT | 66.20 | 67.45 | 65.30 | 64.45 | 61.00 | 63.55 | 64.66 |
| Series (Glg→Swa) | 65.10 | 66.80 | 62.90 | 63.50 | 53.75 | 55.65 | 61.28 |
| Series (Swa→Glg) | 64.10 | **67.35** | 64.55 | **65.45** | 47.50 | 64.20 | 62.19 |
| Parallel | **66.25** | 66.05 | **64.65** | 62.25 | **58.05** | 67.35 | **64.10** |
| Merging | 64.05 | **67.30** | 64.55 | 63.50 | 56.50 | 65.15 | 63.51 |

Table 14: Performance of different LAYRA setups for adding two languages (Galician + Swahili) on XCOPA

| Model | eng | spa | ita | tha | swa | glg | Avg |
|---|---|---|---|---|---|---|---|
| PT | 87.00 | 81.40 | 72.60 | 57.60 | 55.00 | 57.60 | 68.53 |
| Series (Glg→Swa) | 81.00 | 74.40 | 64.20 | 57.40 | 59.40 | **61.80** | 66.37 |
| Series (Swa→Glg) | 83.00 | 76.40 | 62.00 | 57.60 | 60.00 | 59.20 | 66.37 |
| Parallel | 86.00 | 76.80 | 63.20 | 57.00 | **63.40** | 60.80 | 67.87 |
| Merging | **87.00** | **78.80** | **66.40** | **58.00** | 58.20 | 60.20 | **68.10** |

Table 15: Ablation on LAYRA Configurations for Swahili XNLI

| Model | deu | eng | spa | fra | hin | tha | swa | Avg |
|---|---|---|---|---|---|---|---|---|
| PT | 52.05 | 54.90 | 51.33 | 50.12 | 48.96 | 47.39 | 39.24 | 49.14 |
| Swa LoRA | 44.78 | 55.06 | 45.50 | 48.27 | 41.93 | 37.59 | 45.46 | 45.51 |
| Swa LAYRA (1,10) | 49.24 | 53.98 | 46.22 | 48.47 | 44.58 | 43.78 | **47.39** | 47.67 |
| Swa LAYRA (2,10) | 49.92 | 55.42 | 46.63 | **50.64** | 44.98 | 43.94 | 46.39 | 48.27 |
| Swa LAYRA (6,10) | 50.04 | **55.70** | **47.51** | 48.31 | 45.42 | 45.82 | 46.47 | 48.47 |
| Swa LAYRA (10,10) | 49.68 | 54.98 | 46.67 | 48.84 | 44.02 | 46.39 | 47.15 | 48.25 |
| Swa LAYRA (10,6) | **50.36** | 54.74 | 47.15 | 48.47 | **46.91** | 46.83 | 46.83 | **48.76** |
| Swa LAYRA (10,2) | 49.92 | 54.22 | 47.43 | 49.24 | 46.83 | 46.35 | 45.34 | 48.48 |
| Swa LAYRA (10,1) | 49.04 | 54.82 | 48.92 | 48.55 | 45.90 | 45.06 | 46.99 | 48.47 |

Table 16: Ablation on LAYRA Configurations for Swahili PAWS-X

| Model | deu | eng | spa | fra | swa | Avg |
|---|---|---|---|---|---|---|
| PT | 66.20 | 67.45 | 65.30 | 64.45 | 61.00 | 64.88 |
| Swa LoRA | 66.40 | 68.75 | 64.00 | 63.35 | 61.70 | 64.80 |
| Swa LAYRA (1,10) | 65.40 | 69.40 | 61.45 | **64.40** | **64.40** | 65.01 |
| Swa LAYRA (2,10) | 65.00 | 66.70 | 62.60 | 62.75 | 63.00 | 64.01 |
| Swa LAYRA (6,10) | 65.25 | **69.70** | 64.50 | 62.30 | 62.40 | 64.83 |
| Swa LAYRA (10,10) | 66.40 | 68.65 | 63.45 | 63.00 | 62.50 | 64.80 |
| Swa LAYRA (10,6) | 65.10 | **69.70** | **65.35** | 63.40 | 60.45 | 64.80 |
| Swa LAYRA (10,2) | 65.15 | 68.95 | 63.85 | 64.20 | 61.20 | 64.67 |
| Swa LAYRA (10,1) | **66.45** | 69.45 | 64.15 | 64.20 | 63.30 | **65.51** |

Table 17: Ablation on LAYRA Configurations for Swahili XCOPA

| Model | eng | spa | ita | tha | swa | Avg |
|---|---|---|---|---|---|---|
| PT | 87.00 | 81.40 | 72.60 | 57.60 | 55.00 | 70.72 |
| Swa LoRA | 88.00 | **70.60** | **62.00** | 56.60 | **66.20** | **68.68** |
| Swa LAYRA (1,10) | 86.00 | 68.00 | 56.40 | 57.80 | 63.60 | 66.36 |
| Swa LAYRA (2,10) | 85.00 | 69.60 | 59.60 | 57.20 | 64.60 | 67.20 |
| Swa LAYRA (6,10) | **88.00** | 70.00 | 60.80 | 57.20 | 65.00 | 68.20 |
| Swa LAYRA (10,10) | 86.00 | 70.40 | 58.00 | 56.60 | 65.40 | 67.28 |
| Swa LAYRA (10,6) | **88.00** | 69.80 | 59.60 | **58.60** | 66.00 | **68.40** |
| Swa LAYRA (10,2) | 87.00 | 69.60 | 61.20 | 57.80 | 64.60 | 68.04 |
| Swa LAYRA (10,1) | 85.00 | 69.40 | 59.00 | 56.40 | 62.80 | 66.52 |

Table 18: Ablation on LAYRA Configurations for Swahili XStoryCloze

| Model | eng | spa | hin | swa | Avg |
|---|---|---|---|---|---|
| PT | 78.16 | 70.75 | 64.46 | 55.86 | 67.31 |
| Swa LoRA | 76.17 | 66.71 | 63.40 | 65.12 | 67.85 |
| Swa LAYRA (1,10) | 76.17 | 66.51 | 63.34 | 63.07 | 67.27 |
| Swa LAYRA (2,10) | 76.70 | 66.64 | 56.59 | 63.53 | 65.87 |
| Swa LAYRA (4,10) | **77.04** | 66.51 | 62.41 | 64.46 | 67.61 |
| Swa LAYRA (10,10) | **77.04** | **67.97** | 62.41 | **65.06** | **68.12** |
| Swa LAYRA (10,6) | 76.57 | 67.64 | **63.53** | 64.53 | 68.07 |
| Swa LAYRA (10,2) | 76.51 | 66.51 | 63.27 | 63.73 | 67.51 |
| Swa LAYRA (10,1) | 75.58 | 67.57 | 63.20 | 62.94 | 67.32 |

Table 19: LAYRA-SERIES Ablation: Accuracy of varying $\lambda'$ with Galacian adapted model on XNLI

| Model | deu | eng | spa | fra | hin | tha | swa | glg | Avg |
|---|---|---|---|---|---|---|---|---|---|
| Swa 0.0 | 50.84 | **55.81** | 50.64 | 46.22 | 48.07 | 46.43 | **34.66** | 54.02 | 48.34 |
| Swa 0.1 | 49.76 | 54.50 | 51.16 | 47.79 | 47.91 | 44.62 | 37.23 | **55.54** | 48.56 |
| Swa 0.2 | 50.32 | 54.18 | **51.45** | 48.92 | **48.63** | 44.82 | 39.52 | 54.74 | 49.07 |
| Swa 0.3 | 50.16 | 54.58 | 51.37 | 49.60 | 48.31 | 44.54 | 40.92 | 54.32 | 49.23 |
| Swa 0.4 | 49.24 | 54.78 | 51.24 | **49.64** | 48.55 | 44.30 | 42.21 | 53.24 | 49.15 |
| Swa 0.5 | 48.88 | 54.74 | 50.96 | 48.63 | 47.95 | 44.82 | 43.86 | 52.01 | 48.98 |
| Swa 0.6 | 47.55 | 54.30 | 49.92 | 47.59 | 46.67 | 45.14 | 45.06 | 49.69 | 48.24 |
| Swa 0.7 | 46.55 | 54.06 | 48.80 | 46.72 | 45.18 | 45.18 | 46.34 | 47.55 | 47.55 |
| Swa 0.8 | 46.22 | 53.09 | 47.15 | 44.66 | 44.02 | 44.86 | **46.71** | 44.66 | 46.42 |
| Swa 0.9 | 45.34 | 50.88 | 44.18 | 41.08 | 40.68 | 44.74 | 46.59 | 41.92 | 44.43 |
| Swa 1.0 | 44.90 | 48.15 | 42.01 | 40.28 | 40.28 | 44.18 | 45.30 | 37.23 | 42.79 |

Table 20: LAYRA-SERIES Ablation: Accuracy of varying $\lambda'$ with Galician adapted model on PAWS-X

| Model | deu | eng | spa | fra | swa | glg | Avg |
|---|---|---|---|---|---|---|---|
| Swa 0.0 | **67.40** | 67.15 | 64.30 | 63.30 | 50.35 | **65.25** | 62.96 |
| Swa 0.1 | 66.60 | 67.15 | **64.85** | 64.20 | 52.85 | 63.65 | 63.22 |
| Swa 0.2 | 65.95 | 67.10 | 64.50 | 63.45 | 53.80 | 63.10 | **62.98** |
| Swa 0.3 | 65.55 | 67.20 | 63.55 | 63.25 | **54.80** | 61.50 | 62.64 |
| Swa 0.4 | 65.15 | **67.40** | 63.60 | 62.95 | 54.80 | 59.15 | 62.18 |
| Swa 0.5 | 65.10 | 66.80 | 62.90 | 63.50 | 53.75 | 55.65 | 61.28 |
| Swa 0.6 | 65.10 | 65.95 | 62.40 | 61.60 | 52.45 | 51.85 | 59.89 |
| Swa 0.7 | 63.75 | 65.00 | 60.75 | 60.30 | 51.30 | 48.50 | 58.27 |
| Swa 0.8 | 62.40 | 64.25 | 60.70 | 58.80 | 50.80 | 47.10 | 57.34 |
| Swa 0.9 | 61.55 | 63.15 | 58.70 | 56.65 | 50.80 | 46.55 | 56.23 |
| Swa 1.0 | 61.55 | 62.10 | 59.00 | 55.40 | 52.75 | 46.15 | 56.16 |

Table 21: LAYRA-SERIES Ablation: Accuracy of varying $\lambda'$ with Galician adapted model on XCOPA

| Model | eng | spa | ita | tha | swa | glg | Avg |
|---|---|---|---|---|---|---|---|
| Swa 0.0 | **86.00** | 76.80 | 60.20 | 58.40 | 54.40 | **63.20** | 66.50 |
| Swa 0.1 | 83.00 | **78.80** | 61.40 | **58.80** | 56.40 | 62.00 | 66.73 |
| Swa 0.2 | 83.00 | 77.40 | 63.20 | 57.60 | 57.20 | 61.80 | 66.70 |
| Swa 0.3 | **86.00** | 77.00 | **65.20** | 57.60 | 57.00 | 61.20 | **67.33** |
| Swa 0.4 | 83.00 | 76.00 | 64.60 | 57.60 | 58.20 | 61.40 | 66.80 |
| Swa 0.5 | 81.00 | 74.40 | 64.20 | 57.40 | 59.40 | 61.80 | 66.37 |
| Swa 0.6 | 79.00 | 71.60 | 64.00 | 55.80 | 61.80 | 60.00 | 65.37 |
| Swa 0.7 | 78.00 | 69.60 | 62.60 | 55.60 | 62.20 | 59.00 | 64.50 |
| Swa 0.8 | 79.00 | 67.20 | 62.20 | 56.60 | **63.00** | 56.80 | 64.13 |
| Swa 0.9 | 80.00 | 63.80 | 59.20 | 56.80 | 62.80 | 55.40 | 63.00 |
| Swa 1.0 | 78.00 | 62.60 | 56.00 | 57.00 | 62.40 | 55.80 | 61.97 |

Table 22: LAYRA-SERIES Ablation: Accuracy of varying $\lambda'$ with Galician adapted model on XStoryCloze

| Model | eng | spa | hin | swa | glg | Avg |
|---|---|---|---|---|---|---|
| Swa 0.0 | **76.11** | 69.69 | 63.20 | 50.69 | **69.89** | 65.92 |
| Swa 0.1 | 76.37 | **69.82** | 63.27 | 52.22 | 70.28 | 66.39 |
| Swa 0.2 | **76.44** | 69.69 | 64.00 | 54.20 | **70.68** | 67.00 |
| Swa 0.3 | 76.37 | 69.16 | 64.00 | 55.72 | 69.95 | **67.04** |
| Swa 0.4 | 76.17 | 68.83 | **64.46** | 56.92 | 68.70 | 67.02 |
| Swa 0.5 | 75.91 | 68.23 | 64.13 | 58.44 | 67.31 | 66.80 |
| Swa 0.6 | 75.58 | 67.90 | 63.60 | 59.43 | 65.25 | 66.35 |
| Swa 0.7 | 74.98 | 66.98 | 63.80 | 60.75 | 62.94 | 65.89 |
| Swa 0.8 | 74.32 | 65.85 | 64.00 | 61.02 | 61.15 | 65.27 |
| Swa 0.9 | 72.67 | 63.40 | 63.67 | **61.48** | 58.44 | 63.93 |
| Swa 1.0 | 70.62 | 61.68 | 63.53 | 61.48 | 56.19 | 62.70 |

Table 23: Hyperparameters for Experiments

| Hyperparameter | Description | Value |
|---|---|---|
| Epochs | Training epochs | 1 |
| | 1 language: 32 | 32 |
| Batch Size | 2 languages: 64 | 64 |
| | 3 languages: 128 | 128 |
| Sequence Length | Maximum sequence length | 2048 |
| Warm-up Steps | Proportion of total optimization steps | 5% |
| Learning Rate ($\alpha$) | Initial learning rate | 3e-4 |
| Learning Rate Schedule | | Linear |
| Weight Decay ($\lambda$) | Regularization parameter | 0.1 |
| Optimizer | | AdamW |
| Epsilon ($\epsilon$) | Optimizer stability parameter | 1.0e-5 |
| $\beta_1$ | First moment decay rate | 0.9 |
| $\beta_2$ | Second moment decay rate | 0.95 |
| GPUS | Hardware | H100 X2 |

Table 24: New languages Translated Using Google Machine Translate (GMT). All other languages used for our evaluation but not listed here were obtained from the original dataset release.

| Task | Swa | Urd | Glg |
|------|-----|-----|-----|
| XNLI | ✓ | ✓ | ✓ |
| XStoryCloze | ✓ | - | ✓ |
| PAWS | ✗ | ✗ | ✓ |
| XCOPA | ✓ | ✗ | ✓ |
| MGSM | ✓ | ✓ | ✗ |
| MMLU-Lite | ✓ | ✗ | ✗ |

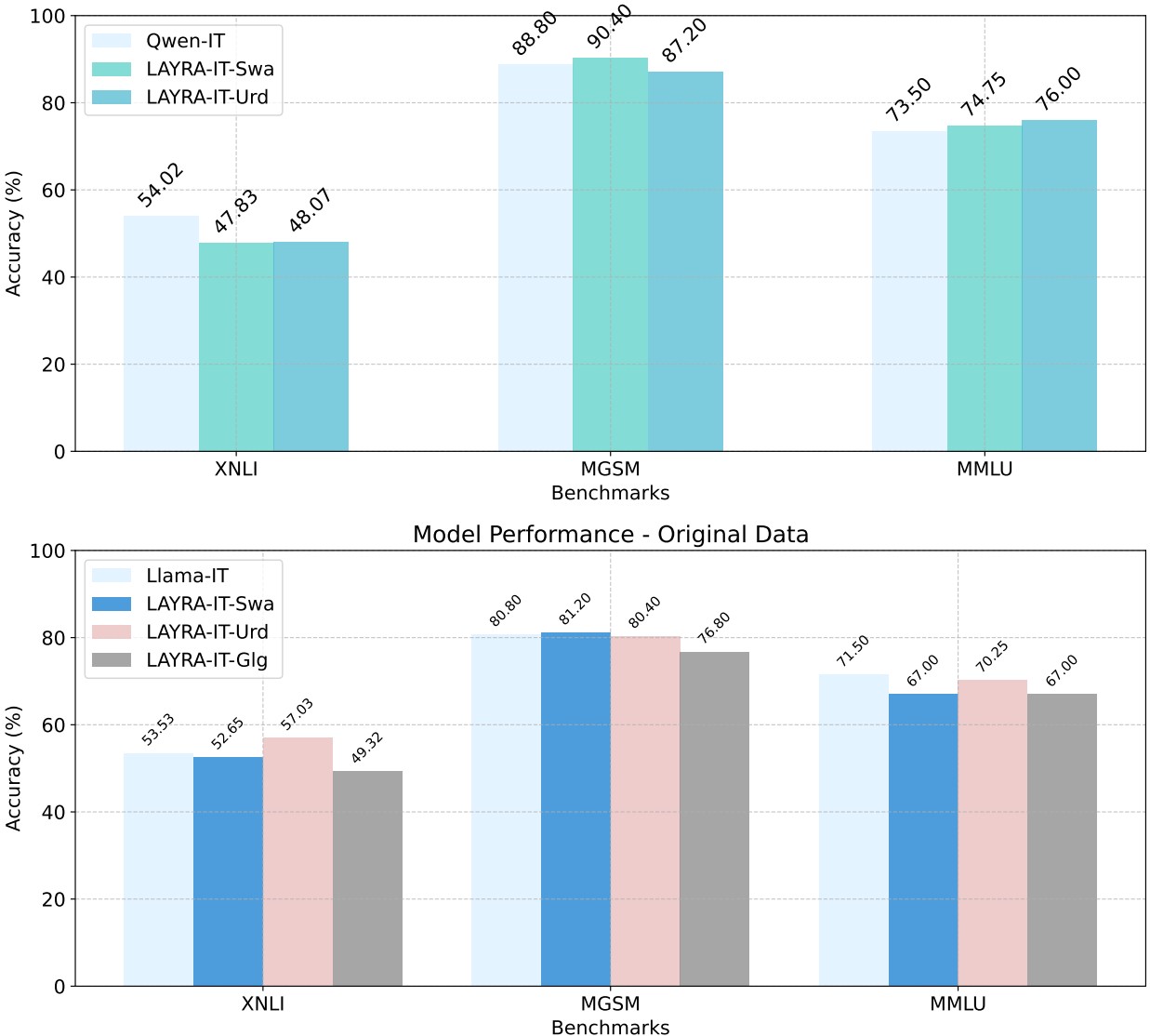

Figure 4: Accuracy of Qwen (top) and Llama (bottom) instruction models vs the LAYRA Instruct on XNLI, MGSM and MMLU for English

