# OpenReview forum: "Continually Adding New Languages to Multilingual Language Models"
_TMLR — Accepted by TMLR_

### Review · Reviewer_jePY · 2025-11-11

**Summary Of Contributions:**

This paper addresses the problem of continually adding new languages to multilingual language models without access to the original pretraining data. The authors propose Layer-Selective LoRA (LayRA), a method that applies Low-Rank Adapters only to the initial and final layers of the transformer. Experiments on adding Galician, Swahili, and Urdu to Llama 3.1 and Qwen 2.5 demonstrate that LayRA effectively balances learning new languages and retaining performance on previously supported ones. The paper also explores sequential language addition and instruction-following transfer via model arithmetic. I recommend acceptance of this paper after revision.

**Additional Comments:**

NA

**Audience:**

Yes

**Audience Explanation:**

The problem of efficiently adapting large language models to new languages is a central challenge in NLP. This paper provides a practical and effective solution that would interest researchers and practitioners working on multilingual models, low-resource languages, and continual learning.

**Claims And Evidence:**

Yes

**Claims Explanation:**

The claims are strongly supported by extensive, well-designed experiments across multiple models, languages, and benchmarks, with the results presented clearly in comprehensive tables and figures that consistently demonstrate the advantages of the proposed method.

**Requested Changes:**

1. Equation (2) contains a subscript inconsistency: it should be $\theta_{N+n+n^\prime}^{\text{series}} = \theta_{N+n} + \lambda^\prime (\phi_{N+n+n^\prime} - \theta_{N+n})$ instead of $\theta_{N+n+n^\prime}^{\text{series}} = \theta_{N+n} + \lambda^\prime (\phi_{N+n+n^\prime} - \theta_{N+n^\prime})$

2. The authors apply LoRA to the earliest 10 and final 2 layers for all experiments, regardless of the base model. However, this setting may be model-dependent, influenced by total depth and training data. Applying the same fixed set of layers to all architectures is suboptimal and may not generalize well.

3. The authors do not explain how they address the issue of adding new tokens when adapting to a language with a script not fully covered by the original model's tokenizer. In practice, this often requires expanding the tokenizer and the model's embedding layer, which introduces architectural changes and complicates parameter-efficient fine-tuning and model merging.

4. The performance of LayRA in a parallel setting (i.e., adding multiple languages simultaneously) is not thoroughly explored, though it is mentioned as an upper bound in Table 2. A direct comparison with parallel training would better contextualize LayRA’s efficiency.

5. Several relevant works on catastrophic forgetting and continual learning are not discussed, such as [1–6]. This limits the positioning of the work within the broader literature.

---

[1] Efficient Test-Time Model Adaptation without Forgetting. ICML 2022.

[2] Test-Time Learning for Large Language Models， ICML 2025.

[3] Efficiently Learning at Test-Time: Active Fine-Tuning of LLMs ，ICLR 2025.

[4] Robust test-time adaptation in dynamic scenarios. CVPR 2023.

[5] Test-Time Training on Nearest Neighbors for Large Language Models，ICLR 2024.

[6] Learning to Generate Gradients for Test-Time Adaptation via Test-Time Training Layers. AAAI 2025.

---

> ### Author Response · Authors · 2025-12-02
>
> We thank the reviewer for their detailed feedback and questions. Please find our responses to your questions and concerns below:
>
>
> > **Equation (2) contains a subscript inconsistency ...**
>
>
>
> We appreciate you pointing out this notation inconsistency. We have corrected it in the revised version of our paper, which we uploaded.
>
> $~$
>
>
> > **The authors apply LoRA to the earliest 10 and final 2 layers for all experiments, regardless of the base model. However, this setting may be model-dependent, influenced by total depth and training data. Applying the same fixed set of layers to all architectures is suboptimal and may not generalize well.**
>
> We would like to clarify that our method for layer selection is *not* based on a fixed heuristic but a ratio-based selection established from our ablation experiments in Appendix A.2. We used 10 early layers and 2 final layers for both Llama-3 and Qwen because both models coincidentally have a similar number of layers, being 32, making the ratio identical across architectures. For models with a different depth (e.g., 40 layers), this same ratio would translate to selecting 12 early layers and 3 final layers, corresponding to approximately 30% and 7% of the total depth, respectively.
>
> $~$
> > **The authors do not explain how they address the issue of adding new tokens when adapting to a language with a script not fully covered by the original model's tokenizer. In practice, this often requires expanding the tokenizer and the model's embedding layer, which introduces architectural changes and complicates parameter-efficient fine-tuning and model merging.**
>
> We would like to clarify that we did not explore the addition of new tokens in our work. Expanding the tokenizer would increase parameter overhead and alter the tokenizer’s merging rules. Any modification to these merging rules typically leads to catastrophic forgetting, making this approach incompatible with our objective of learning a new language without forgetting the old ones. This tokenizer mismatch issue is only dominant in the Llama tokenizer because Urdu text is less represented in the model’s training data. The Qwen tokenizer, however, exhibits less of this issue as it was trained on Arabic text.
>
> $~$
>
>
> > **The performance of LayRA in a parallel setting (i.e., adding multiple languages simultaneously) is not thoroughly explored, though it is mentioned as an upper bound in Table 2. A direct comparison with parallel training would better contextualize LayRA’s efficiency.**
>
> We appreciate the reviewer’s point regarding parallel adaptation. Our work is intentionally scoped around the single-language addition setting, which is the dominant practical scenario for multilingual model extension in industry and in most prior research (Sani et al., 2025).
>
> $~$
>
> > **Several relevant works on catastrophic forgetting and continual learning are not discussed, such as [1–6]. This limits the positioning of the work within the broader literature.**
> We thank the reviewer for highlighting these related works. All the suggested papers focus on Test-Time Adaptation (TTA), where the goal is to adapt a model to each test instance rather than mitigate catastrophic forgetting during finetuning. Among these works, the subset that addresses catastrophic forgetting primarily relies on LoRA finetuning as their main method. We have shown the results of LoRA finetuning in all our experiments, and we will cite the most relevant of these works in our revised version.
>
> $~$
> ### References
> [1] Sani, Samin Mahdizadeh, et al. "Extending LLMs to new languages: A case study of llama and Persian adaptation." Proceedings of the 31st International Conference on Computational Linguistics. 2025.
> .

---

### Review · Reviewer_YJQc · 2025-11-25

**Summary Of Contributions:**

This paper studies how to add new languages to multilingual LLMs without access to original pretraining data, a setting where standard continued pretraining causes catastrophic forgetting. It introduces LayRA, a method that applies LoRA only to the early and late transformer layers. After training, the method performs a model-arithmetic merge, adding only a scaled version of the learned “language vector,” which dramatically reduces forgetting.

**Additional Comments:**

- it's a a bit concerning that, when we are at the end of 2025, there are only two technical papers from 2025 being discussed (other than Gemini report). It's a bit hard to believe that the discussion is comprehensive given the progress these days. (although I don't have a list of papers for the authors to add.)

**Audience:**

Yes

**Audience Explanation:**

- The paper builds upon a very intuitive and working observation: multi-lingual LLM mostly use English to reasoning in the middle layers, and use the remaining layers to process different languages, the proposed idea also naturally align with this observation.

- The paper has a good range of ablation studies to verify the claims.

**Broader Impact Concerns:**

None.

**Claims And Evidence:**

Yes

**Claims Explanation:**

The paper has fairly reasonable scope of ablation studies, most of the claims are grounded.

**Requested Changes:**

- The layer selection seems to be mostly heuristic, while it is reasonable to test it empirically, it does not seem to consider whether the selected setting will be the best for future languages. In particular, it is also worth reporting whether different languages benefit from different layer settings.

- The final evaluation in Table 1 does not so convincingly favor the proposed method. If we only look at the "average" column, there are six numbers favoring LayRA, and five numbers favoring LoRA, this seems to undermine the core contribution of this paper.

- The name LayRA is a bit misleading, the core contribution seems to be about the "merge", but this name indicates it's an improvement of Lora.

---

> ### Author Response · Authors · 2025-12-02
>
> We sincerely appreciate the reviewer for their detailed feedback and questions. Please find our responses to your questions and concerns below:
>
> $~$
>
> > **The layer selection seems to be mostly heuristic. While it is reasonable to test it empirically, it does not seem to consider whether the selected setting will be the best for future languages. In particular, it is also worth reporting whether different languages benefit from different layer settings.**
>
> Our layer-selection strategy is not arbitrary but is grounded in prior work (Zhao et al., 2024; Wendler et al., 2024) that provides extensive ablations showing that multilingual models process languages within their layers in a particular order. Based on this evidence and our ablation experiments in Appendix A.2, we adopt a language-agnostic selection rule rather than tuning layers for each individual language. A language-specific configuration would contradict these findings and would also make the method impractical to apply when adding multiple languages either sequentially or in parallel. We have clarified this motivation in the revised version.
>
> $~$
>
> > **The final evaluation in Table 1 does not so convincingly favor the proposed method. If we only look at the "average" column, there are six numbers favoring LayRA, and five numbers favoring LoRA, this seems to undermine the core contribution of this paper.**
>
> We would like to clarify that in Table 1, LayRA actually outperforms other methods with a win ratio of **7:4**, not **6:5** as stated by the reviewer. For transparency, we have expanded the average column in the tables below to clearly show this comparison.
>
> ### XNLI
> | **Model**       | **Swa**    | **Urd**    | **Glg**    |
> |-------------|--------|--------|--------|
> | PT          | 39.24  |  36.43        | 47.57         |
> |-------------|--------|--------|--------|
> | Full        | 40.95  | 37.57  | 41.21  |
> | Layer-sel.  | 43.91  | 39.34  | 43.40  |
> | LoRA        | **46.82**  | 44.82  | 46.49  |
> | LAYRA       | 46.64  | **45.58**  | **47.14** |
>
> $~$
> ### PAWS-X
> | **Model**       | **Swa**    | **Urd**    | **Glg**    |
> |-------------|--------|--------|--------|
> | PT          | 61.00  |  54.25      |    63.55    |
> |-------------|--------|--------|--------|
> | Full        | 59.36  | 54.11  | 57.37  |
> | Layer-sel.  | 59.82  | 56.10  | 58.57  |
> | LoRA        | 62.97  | 59.58  | 60.89  |
> | LAYRA       | **63.25** | **60.79** | **61.03** |
>
> $~$
> ### XCOPA
> | **Model**       | **Swa**    | **Urd**    | **Glg**    |
> |-------------|--------|--------|--------|
> | PT          | 55.00  |  58.80     |  57.60      |
> |-------------|--------|--------|--------|
> | Full        | 58.97  | 56.09  | 60.09  |
> | Layer-sel.  | 61.00  | 58.37  | 62.34 |
> | LoRA        | **64.74**  | 65.57  | 65.14  |
> | LAYRA       | 64.09  | **66.26** | **65.20** |
> $~$
> ### XStoryCloze
> | **Model**       | **Swa**    | **Urd**    | **Glg**    |
> |-------------|--------|-----|--------|
> | PT          |55.86  |  -   |    64.46    |
> |-------------|--------|-----|--------|
> | Full        | 57.56  | —   | 60.07  |
> | Layer-sel.  | 63.51  | —   | 63.65 |
> | LoRA        |  **65.86**  | —   | **66.50** |
> | LAYRA       | 65.59  | —   | 65.92  |
>
> $~$
>
> > **The name LayRA is a bit misleading, the core contribution seems to be about the "merge", but this name indicates it's an improvement of Lora.**
>
> We would like to mention that our core contribution is about language addition and instruction following without catastrophic forgetting. Our main method is **Lay**er Selective Lo**RA**, which was how we derived our name.
>
> $~$
>
> > **It's a bit concerning that, when we are at the end of 2025, there are only two technical papers from 2025 being discussed (other than Gemini report). It's a bit hard to believe that the discussion is comprehensive given the progress these days. (although I don't have a list of papers for the authors to add.)**
>
> It is true that more open source models have been released in 2025; however, not many of them are multilingual models. This underscores the importance of our work; speakers of non-English languages are neglected in the current AI boom.   We have updated our citations to include additional multilingual models released in 2025, including Qwen 3, Llama 4, Command A, Mistral, and Babel (a massively multilingual LLM), to ensure our discussion is comprehensive.
>
> $~$
> ### References
> [1] Zhao, Yiran, et al. "How do large language models handle multilingualism?." Advances in Neural Information Processing Systems 37 (2024): 15296-15319.
> [2] Wendler, Chris, et al. "Do llamas work in english? on the latent language of multilingual transformers." Proceedings of the 62nd Annual Meeting of the Association for Computational Linguistics (Volume 1: Long Papers). 2024.

---

### Review · Reviewer_Sjpy · 2025-11-26

**Summary Of Contributions:**

### Main contributions

- A replay-free continual learning framework for incrementally adding new languages to pretrained multilingual LMs
- LayRA (Layer-selective LoRA): a parameter-efficient method that applies LoRA adapters only to selected initial and final transformer layers while freezing middle layers
- Two strategies for sequential language addition: LayRA-series (iterative) and LayRA-merge (parallel adaptation + merging)
- A model arithmetic approach to enable instruction-following in new languages without target-language instruction data

### Key strengths
- Well-written and clearly presented with comprehensive experimental evaluation across multiple languages (Galician, Swahili, Urdu), benchmarks (XNLI, PAWS-X, XCOPA, XStoryCloze), and model families (Llama 3.1, Qwen 2.5)
- Conceptually motivated approach based on insights about layer-wise language processing in multilingual models
- Demonstrates learning-retention tradeoffs that may be favorable to full CPT and standard LoRA

### Key weaknesses
- Limited scale: experiments use only 1.2B tokens per language, which is orders of magnitude smaller than realistic pretraining scenarios (100B+ tokens), raising questions about scalability of findings
- Unclear practical motivation: the sequential language addition scenario may be less common than batch addition or adding a single language in practice.
- Inconsistent and sometimes counterintuitive results that require better explanation
- Limited evaluation of sequential adaptation (only 2-stage with specific language pairs), making it difficult to assess generalization

**Audience:**

Yes

**Audience Explanation:**

This paper addresses a practical problem, potentially faced by researchers working with open multilingual models: how to extend language coverage when original pretraining data is unavailable, which is increasingly common in the current LLM research landscape. The continual learning methodology and findings are relevant to the broader machine learning community working on forgetting and retention dynamics, parameter-efficient fine-tuning, and model merging techniques. Additionally, the insights about layer-wise language processing and selective adaptation could inform future work on efficient multilingual model development.

**Claims And Evidence:**

Yes

**Claims Explanation:**

Overall, the experiments are somewhat convincing. The authors systematically compared LayRA against meaningful baselines (full CPT, LoRA CPT, layer-selective CPT) across diverse language pairs that vary in script and linguistic similarity to the base model's training corpora. The results consistently show LayRA achieving good retention-learning tradeoffs across multiple evaluation benchmarks and two different model families (Llama 3.1 and Qwen 2.5). The ablations are comprehensive enough and we can clearly see how findings change with different hyperparameter settings.

Yet, I would like to raise some concerns and ask questions to the authors concerning the interpretations of the results:

- There appears to be a mismatch between experimental setup descriptions and hyperparameter discussions. In Section 3.2's first paragraph, the authors mention determining λ′ = 0.5 for LayRA-series when discussing single-language addition, but LayRA-series (Equation 2) involves n and n′ languages, suggesting it corresponds to the two-stage setup. Is there a mistake in my understanding?
- The notation in Equation 1 should be explained more clearly—specifically, how this equation accommodates both full CPT (where $\theta$ and $\phi$ have same dimensionality) and LoRA (where dimensions differ).
- According to Table 1, pretrained model already achieves better average results than all continually pretrained models. Is my understanding correct? If that's the case, do we conclude that CPT is actually detrimental? How would the authors interpret this?
- Full CPT sometimes decreases performance on the target language (e.g., Urdu + PAWS-X in Table 1), which is unexpected. This requires discussion about potential causes—is this due to insufficient training, data quality, or other factors?
- The claim about Urdu's script being poorly represented in Llama 3.1's tokenizer would benefit from supporting evidence or citations.
- I am a little skeptical about the conclusion that "Galician-trained model improves English performance". I'm not sure if this occurs consistently or it's just a coincidence as Llama3.1 was trained on trillions of tokens anyway. Also, wouldn't we expect performance to increase in Spanish as well, which is not the case if I read Table 1 right?
- The results on "adding instruction tuning" are rather inconsistent, which is understanble. One interesting comparison would be to test original Qwen and Llama on the same questions asked in English. This would give an upper bound on the overall performance and help us identify how effectively these new languages were acquired.

**Requested Changes:**

A discussion on my above questions/comments would very much strengthen the paper. Also, I have some minor comments:

- Figure 1: Please add brief explanations of LayRA-series and LayRA-merge directly in the figure or caption for clarity.
- Equation 1: Explain how this equation accommodates both full CPT and LoRA. More specifically, if $\phi$ and $\theta$ are of the same dimensionality, then the equation cannot stand for LoRA as LoRA requires new parameters.
- Why are there two different symbols (theta and phi) for network parameters?
- Terminology: Please define "language vector" when first introduced, and clarify whether training in Equation 1 uses all $n$ languages or a single language.
- Motivation: Please provide explicit justification for why CPT is a realistic setup, aka, the model is not trained on all $N$ original languages.
- Figure 2 is very cool, showing the tradeoffs nicely. Maybe repeat it for Llama results?
- It would be great to have language-specific result tables that reflects the findings in "Impact of language relatedness and scripts" paragraph.

---

> ### Author Response · Authors · 2025-12-02
>
> We sincerely appreciate the reviewer for their detailed feedback and questions. Please find our responses to your questions and concerns below:
>
> $~$
> >**Limited scale: experiments use only 1.2B tokens per language, which is orders of magnitude smaller than realistic pretraining scenarios (100B+ tokens), raising questions about scalability of findings**
>
> We acknowledge that the current model pretraining paradigm scales pretraining to over 100B tokens; however, in low-resource settings, it is practically impossible to get high-quality datasets at this scale. We used Fineweb2 for our experiment, and this dataset was prepared and well-curated by the HuggingFace team. Even with all the resources at their disposal, they were only able to gather 1.2B tokens for Swahili. We also explored the WURA dataset (Oladipo et al., 2023), the only available pretraining alternative, but we found the quality of the dataset to lead to worse performance after CPT.
>
> $~$
> > **Unclear practical motivation: the sequential language addition scenario may be less common than batch addition or adding a single language in practice.**
>
>
> We understand the reviewer’s concern about our LayRA-Series setup. We particularly designed this experiment to simulate common practices as seen in existing works that have explored the continual model merging problem for adding new skills to a model (Tang et al., 2025; Shenaj et al., 2025;  Dziadzio et al., 2025; Phan et al., 2025). For us, we design this as a continual model merging problem for adding new languages to an LLM.
>
> $~$
>
> >**Limited evaluation of sequential adaptation (only 2-stage with specific language pairs), making it difficult to assess generalization.**
>
> The main focus of our paper was to explore multiple ways of adding new languages to an already pretrained model and make it follow instructions in the new language after the language addition. For sequential addition, we performed 4 different finetuning experiments with 2 languages, and we evaluated on 4 benchmarking datasets. We believe these experiments fully evaluate our exploration of sequential language addition, given our academic compute budget.
>
> $~$
>
> >**There appears to be a mismatch between experimental setup descriptions and hyperparameter discussions. In Section 3.2's first paragraph, the authors mention determining λ′ = 0.5 for LayRA-series when discussing single-language addition, but LayRA-series (Equation 2) involves n and n′ languages, suggesting it corresponds to the two-stage setup. Is there a mistake in my understanding?**
>
> In that paragraph, we mean that, for adding a single language, we set $\lambda=1$. Afterwards, if we desire to add more languages to this new model, we will set $\lambda’ =0.5$. We separate $\lambda$ and $\lambda'$ because the first language addition uses a different lambda value from the second language addition. We have clarified this better in our revised submission.
>
> $~$
>
> > **The notation in Equation 1 should be explained more clearly—specifically, how this equation accommodates both full CPT (where  θ and Φ and have same dimensionality) and LoRA (where dimensions differ).**
>
> Both θ and Φ have the same dimensions. Although the added LoRA matrix may seem to introduce more parameters, we merge the LoRA weights back into the main model after training. This ensures that θ and Φ have the same parameters. We have clarified in our revised submission.
>
> $~$
> > **Inconsistent and sometimes counterintuitive results that require better explanation**
> > **According to Table 1, pretrained model already achieves better average results than all continually pretrained models. Is my understanding correct?**
>
> We believe there may be a misunderstanding of Table 1, as this is not the case. We combine the results of 3 different languages and models in Table 1 due to space constraints. Each cell represents the result from 3 different models training on Swahili, Urdu, and Galician, separated by “/’ Table 1 shows that CPT always outperforms the pretrained model for the language it was trained on, but experiences extreme catastrophic forgetting on languages it was not trained on. Please see Tables 3, 4, 5, and 6 for the expanded results of Table 1.
>
> $~$
>
> >**Full CPT sometimes decreases performance on the target language (e.g., Urdu + PAWS-X in Table 1), which is unexpected. This requires discussion about potential causes—is this due to insufficient training, data quality, or other factors?**
>
> We note that this occurs in only 1 out of 11 results in our evaluations. We believe the drop may be due to a shift in training data distribution, where the original pretraining corpus is better aligned with PAWS-X than the CPT data, leading to a slight degradation.

---

> ### Author Response · Authors · 2025-12-02
>
> $~$
> > **The claim about Urdu's script being poorly represented in Llama 3.1's tokenizer would benefit from supporting evidence or citations.**
>
> Our statement is based on the fact that Llama 3.1 was not trained on Urdu. Since Urdu uses a non-Latin script, the tokenizer naturally lacks adequate coverage of its characters, leading to poor representation. We performed an analysis of the tokenization pattern to verify this, and we found that the Llama tokenizer produces a lot of invalid tokens compared to the Qwen tokenizer. We highlighted this already in our paper, where we said that “the Qwen 2.5 models trained with Urdu exhibit a higher retention rate”.
>
> | **Tokenizer** | **Text (English/Urdu)** | **Tokens Count** |
> |--------------|------------------|------------------|
> | **English** | We thank the reviewers for their kind feedback. | 9 (including the period) |
> | **Llama** | '�', '�', 'م', ' ج', 'ائز', '�', '�', ' ل', 'ین', '�', '�', ' وال', 'و', '�', '�', ' ک', '�', '�', ' ان', ' ک', '�', '�', ' ا', 'چ', 'ھ', '�', '�', ' تاث', 'رات', ' ک', '�', '�', ' لی', '�', '�', ' شک', 'ری', '�', '�', ' اد', 'ا', ' ک', 'رت', '�', '�', ' ', '�', '�', 'ی', '�', '�', '�', '�' | 54 |
> | **Qwen** | 'ہ', 'م', ' ج', 'ائز', 'ہ', ' ل', 'ین', 'ے', ' وال', 'و', 'ں', ' ک', 'ے', ' ان', ' ک', 'ے', ' ا', 'چ', 'ھ', 'ے', ' ت', 'اث', 'رات', ' ک', 'ے', ' ل', 'ی', 'ے', ' ش', 'ک', 'ر', 'ی', 'ہ', ' ا', 'دا', ' ک', 'رت', 'ے', ' ', 'ہ', 'ی', 'ں', '۔' | 43 |
>
>
> > **I am a little skeptical about the conclusion that "Galician-trained model improves English performance". I'm not sure if this occurs consistently or it's just a coincidence as Llama3.1 was trained on trillions of tokens anyway. Also, wouldn't we expect performance to increase in Spanish as well, which is not the case if I read Table 1 right?**
>
> We agree that this effect may be coincidental in our results. One possible explanation is that Llama 3.1 was predominantly trained on English, so even small distributional shifts introduced during finetuning can more easily be observed in English performance compared to other languages. This may also explain why the pattern does not consistently extend to Spanish.
>
> $~$
> > **The results on "adding instruction tuning" are rather inconsistent, which is understanble. One interesting comparison would be to test original Qwen and Llama on the same questions asked in English....**
>
> We thank the reviewer for this suggestion. This comparison is already included in our paper. Please see Figure 5, where we evaluate the original Qwen and Llama on three benchmarks. We observe that LayRA sometimes even improves the original model’s performance on English.
>
> $~$
>
> > **Figure 1: Please add brief explanations of LayRA-series and LayRA-merge directly in the figure or caption for clarity.**
>
> We have added this to our uploaded revised version.
>
> $~$
> > **Why are there two different symbols (theta and phi) for network parameters?**
>
> We used theta ($\theta$) to represent the pretrained model and phi ($\phi$) to represent the model obtained after adaptation using either CPT, Layer-selective finetuning, LoRA, or LyRA.
>
>
> $~$
> > **Terminology: Please define "language vector" when first introduced, and clarify whether training in Equation 1 uses all n languages or a single language.**
>
> We appreciate this comment and have made the necessary changes in our paper. Particularly, we clarified that training could use either a single language or multiple languages. This is the case because Equation 1 is a generalized version of our problem statement.
>
>
> $~$
> > **Motivation: Please provide explicit justification for why CPT is a realistic setup, aka, the model is not trained on all N original languages.**
>
> We added this justification to the introduction of our paper already. Particularly, we mentioned that “As resources for new languages become available, continually supporting them in such models is not trivial. Retraining them from scratch is often prohibitively expensive, so practitioners typically adopt an incremental continued pretraining (CPT) strategy to incorporate new languages (Csaki et al., 2023). However, it often results in catastrophic forgetting of previously supported languages.”

---

> > ### Author Response · Authors · 2025-12-02
> >
> > $~$
> > > **Figure 2 is very cool, showing the tradeoffs nicely. Maybe repeat it for Llama results?**
> >
> > We thank the reviewer for the kind comment and have added a similar plot for the Llama results. Please see our Figure 4 in Appendix A1.
> >
> > $~$
> > > **It would be great to have language-specific result tables that reflects the findings in "Impact of language relatedness and scripts" paragraph.**
> >
> > We would like to note that this paragraph aggregates insights from multiple results across the paper. We considered creating a dedicated table, but found that it introduced substantial repetition. Instead, we added references to the specific tables from which these insights are drawn.
> >
> >
> > $~$
> > ### **References**
> > [1] Tang, Anke, et al. "Merging models on the fly without retraining: A sequential approach to scalable continual model merging." arXiv preprint arXiv:2501.09522 (2025).
> >
> > [2] Shenaj, Donald, et al. "K-Merge: Online Continual Merging of Adapters for On-device Large Language Models." arXiv preprint arXiv:2510.13537 (2025).
> >
> > [3] Dziadzio, Sebastian, et al. "How to Merge Your Multimodal Models Over Time?." Proceedings of the Computer Vision and Pattern Recognition Conference. 2025.
> >
> > [4] Phan, Hoang, et al. "Toward a Holistic Approach to Continual Model Merging." arXiv preprint arXiv:2509.23592 (2025).
> >
> > [5]. Oladipo, Akintunde, et al. "Better quality pre-training data and t5 models for African languages." Proceedings of the 2023 Conference on Empirical Methods in Natural Language Processing. 2023.
> > [6] Chung, Hyung Won, et al. "Scaling instruction-finetuned language models." Journal of Machine Learning Research 25.70 (2024): 1-53.

---

### Decision · Action_Editor_PN2e · 2026-02-02

**Recommendation:** Accept with minor revision

**Additional Comments:**

After reviewing the submission in detail, the Action Editor (AE) requests the following minor revisions / that the following questions be answered in the manuscript:

1. Are the authors certain that the pretraining data of Qwen did not include Galician, Swahili, Urdu? The authors should explicitly state the assumption they are making or the knowledge they have about this in the paper.
2. The choice of layers for LayRA appears somewhat ad hoc, as described in the paper: "these layers were empirically determined to give the best average performance between retention and learning." As Reviewer YJQc notes, the authors do not provide details of their validation procedure (Appendix A.2 is extremely terse). The authors should either attest to the generalizability of this design choice (e.g., was it validated on an additional held-out set of data?) or explicitly acknowledge the lack of further exploration as a limitation of the work.
3. The following passage on p. 6 should be further explained: "Not all languages with which we experiment have datasets available for all tasks. For languages for which datasets are not available, we translate the English subset of the task to the missing language using Google Machine Translate (March 2025) (see Table 24 in Appendix A.4 for languages that required translations). We perform qualitative analysis to ensure that the translations are accurate." Specifically, the authors should describe what the qualitative analysis consisted of. It should be reproducible.
4. Per the discussion with Reviewer Sjpy, the authors should address the lack of grounds for the speculation about improved performance on Romance pretraining languages, which does not uniformly apply (e.g., to Spanish). More generally, the authors should do a better job of distinguishing conjecture from evidenced claims throughout the paper. For example, the rebuttal to Reviewer Sjpy states: "We note that this occurs in only 1 out of 11 results in our evaluations. We believe the drop may be due to a shift in training data distribution, where the original pretraining corpus is better aligned with PAWS-X than the CPT data, leading to a slight degradation." This is entirely conjecture and should either be removed or explicitly stated as such.
5. Address the requested changes of Reviewer Sjpy. I especially recommend moving the added Figure 4 into the main text, as it is easier to parse than the table of results.
6. The appendices under Section A should refer to specific tables; at present, they read as little more than dangling captions.
7. The caption of Figure 2 reads "Average accuracy of learning vs retention (x, y)." Canonically, "y vs. x" is the standard convention, whereas the figure uses the opposite ordering. Please correct.
8. The authors should release the annotated data as promised in the paper.

**Audience:**

Yes

**Audience Explanation:**

The paper addresses an application domain (replay-free continual learning of new languages for LLMs) that is likely of interest to a subset of the TMLR community even if it is not the most common application setting for machine translation (cf. retraining). Reviewer Sjpy raised concerns about the realism of the continual language learning setting, but I assess that the authors have sufficiently motivated this setting as worthy of study even if it is not in wide use and may differ exactly from how any one lab implements continual pre-training. The other reviewers recommended the paper for acceptance on the basis that the setting is interesting, the results represent a modest improvement, and the methodology is appropriate, though it does not constitute a significantly novel approach. I concur with this assessment and similarly recommend acceptance to TMLR, subject to the minor revisions requested below.

**Claims And Evidence:**

Yes

**Claims Explanation:**

The paper addresses replay-free continual learning of new languages and builds on two conjectures: (1) that LoRA reduces forgetting, and (2) that the cascade of language processing in LLMs follows a structure of source-language encoding, English "reasoning," and target-language decoding. Building on these conjectures, the paper proposes LayRA, which demonstrates improved performance for continual learning of new languages and enables model arithmetic that improves instruction-following ability. The claims are supported by empirical evidence, though the improvements are modest.

One reviewer (YJQc) weakly recommended rejection on the basis that a win ratio of 7:4 in improvements to non-improvements in continual language learning is not significant ("6:5 is not a big change from 7:4"). However, at TMLR results should be evaluated in the context of the claims of the work, not by decontextualized empirical standards of any specific reviewer. As these modest results are communicated appropriately by the authors, and there are other contributions to the work, I find overall the evidence adequate to support the paper's claims. Last, Reviewer Sjpy raised technical concerns that were adequately addressed in the author response.